# High resolution long-read telomere sequencing reveals dynamic mechanisms in aging and cancer

Tobias T. Schmidt[1,4], Carly Tyer[2,4], Preeyesh Rughani[2,4], Candy Haggblom[1], Jeffrey R. Jones [1], Xiaoguang Dai[2], Kelly A. Frazer[3], Fred H. Gage [1], Sissel Juul [2], Scott Hickey [2,5] ✉ & Jan Karlseder [1,5] ✉

Telomeres are the protective nucleoprotein structures at the end of linear eukaryotic chromosomes. Telomeres' repetitive nature and length have traditionally challenged the precise assessment of the composition and length of individual human telomeres. Here, we present Telo-seq to resolve bulk, chromosome arm-specific and allele-specific human telomere lengths using Oxford Nanopore Technologies' native long-read sequencing. Telo-seq resolves telomere shortening in five population doubling increments and reveals intrasample, chromosome arm-specific, allele-specific telomere length heterogeneity. Telo-seq can reliably discriminate between telomerase- and ALT-positive cancer cell lines. Thus, Telo-seq is a tool to study telomere biology during development, aging, and cancer at unprecedented resolution.

Mammalian telomeres, the nucleoprotein structures at the end of eukaryotic linear chromosomes, consist of 5'-TTAGGG-3' repeats and terminate in a single-stranded G-rich overhang[1,2]. The overhang can fold back and form a telomeric loop (T-loop)[3]. The hexameric protein complex shelterin binds telomeric repeats and stabilizes the T-loop[4]. Together, the T-loop[5,6] and shelterin[4,7] protect the telomeric chromosome ends from being recognized as DNA double-strand breaks[8]. Due to the "end-replication problem" and subsequent processing, telomeres in human somatic cells shorten with every round of DNA replication[1,9,10]. Short, deprotected telomeres are recognized by the DNA damage response, and either trigger a permanent cell cycle arrest named replicative senescence[11] or, in cells deficient for the p53 and Rb checkpoint pathways, replicative crisis[12–14], a state with extensive innate immunity-driven, autophagy-dependent cell death[15,16]. Both replicative senescence and crisis restrict the maximum number of cell divisions of human somatic cells and act as powerful, telomere-dependent proliferation barriers against human carcinogenesis[17]. Thus, to overcome proliferation barriers and acquire replicative immortality, cancer cells must activate a telomere maintenance mechanism (TMM)[18]. Whereas most cancers reactivate the reverse transcriptase telomerase (TERT)[19,20], 10–15% of all cancers maintain their telomeres by the recombination-based "alternative lengthening of telomeres" (ALT) mechanism[21]. The absence of TMM in human somatic cells makes TMM-specific vulnerabilities attractive targets for personalized cancer therapy[22]; however, it is critical to efficiently distinguish between ALT[+] and TERT[+] cancers, which is currently challenging in the clinical setting.

To measure telomere length, various methods have been developed, including terminal restriction fragment (TRF) analysis[9], STELA[23], TeSLA[24], quantitative PCR[25], Q-FISH[26], flow FISH[27,28], DNA combing[29], and telomere length estimates based on next-generation sequencing data[30]. These methods use either telomere enrichment, staining of telomeres with specific probes, or a combination of both. However, these traditional methods fail to resolve chromosome arm and allele-specific composition of individual telomeres due to their repetitive nature and length. With the advent of DNA long-read sequencing, it is now possible to sequence entire telomeres and harvest subtelomeric information to annotate individual telomeric reads to specific chromosome arms. In the budding yeast *Saccharomyces cerevisiae*, chromosome arm-specific telomere length and telomere shortening was

[1]Salk Institute for Biological Studies, La Jolla, CA 92037, USA. [2]Oxford Nanopore Technologies, Inc., New York, NY, USA. [3]Institute of Genomic Medicine, University of California, San Diego, La Jolla, CA 92093-0761, USA. [4]These authors contributed equally: Tobias T. Schmidt, Carly Tyer, Preeyesh Rughani. [5]These authors jointly supervised this work: Scott Hickey, and Jan Karlseder. ✉e-mail: scott.hickey@nanoporetech.com; karlseder@salk.edu

resolved using Oxford Nanopore Technologies long-read sequencing[31]. Further, nanopore long-read sequencing was recently applied to measure telomere shortening in an RTEL1 mutant mouse model and to compare the mouse to the human telomere length[32]. For human telomeres, a recent report combined a telomeric pulldown with restriction enzyme digest and PacBio HiFi long-read sequencing to measure telomere length and telomere variant repeats in cultured human cells and patient cells[33]. However, due to the protocol's stringent restriction enzyme digest, the annotation of telomeric reads to chromosome arms was only possible for nine chromosome arms, and the very long telomeres present in ALT[+] cells[21] are incompatible with the processivity of the PacBio HiFi DNA polymerase[33]. Furthermore, data from the "telomere-to-telomere" (T2T)[34,35] and the Genome in the Bottle Consortiums[36] as well as a twin study[37] demonstrated that human whole-genome long-read sequencing can be utilized to analyze human telomere length and composition. However, as the telomeric content of human diploid cells is approximately only 0.015% of the total genome, telomere length measurements based on whole-genome long-read sequencing are inefficient, and telomere enrichment is necessary. Here, we developed Telo-seq to efficiently sequence entire human telomeres using nanopore sequencing and applied it to explore bulk, chromosome arm, and allele-specific human telomere length and composition in aging and cancer.

## Results

### Telo-seq

To enrich for telomeres, telorette-based telomere adapters[23,24] were first annealed to the G-overhang and ligated to the C-strand (Fig. 1a). Next, genomic DNA was digested with the blunt-end restriction enzyme EcoRV. To reduce concatemer ligation, a dA-tail was added to the blunt ends prior to splint adapter annealing and sequencing adapter ligation. After nanopore sequencing, bases were called using a customized Bonito telomere model (Supplementary Fig. 1a). Next, the reads were filtered for quality, the telomeric motif was identified, and reads were filtered for expected structure (Supplementary Fig. 1a, b). To annotate reads to individual chromosome arms, reads were mapped to a collection of well-annotated subtelomeric sequences[34,38] (Supplementary Fig. 1c).

To evaluate Telo-seq, we used the B-lymphocyte cell line HG002, which was sequenced previously as part of the Human Pangenome Reference and Genome in a Bottle Consortiums and a high-quality telomere-to-telomere assembly is publicly available[38,39]. First, we compared telomere enrichments of Telo-seq to long-read sequencing of high-molecular-weight DNA and AluI/MboI double-digested genomic DNA (the restriction enzymes used in traditional TRF analysis[16,40]). Telo-seq resulted in the most telomeric reads of all tested protocols and increased telomeric reads 46-fold relative to whole-genome sequencing (Supplementary Fig. 1d, e, g). Second, we compared the Telo-seq bulk telomere length measurement of two independent HG002 replicates with TRF. Based on TRF analysis, HG002 cells have a bulk telomere length of 4 to 6 kilobases (kb) (Fig. 1b and Supplementary Fig. 1f). In line with the TRF results, Telo-seq revealed a mean telomere length of 5247 and 5270 base pairs (bp) for each independent replicate, with a standard deviation of 2085 and 2135 bp, respectively (Fig. 1c, d, Supplementary Fig. 1f, and Supplementary Table 1). Furthermore, both replicates showed very similar subtelomere and telomere length distributions (Fig. 1c, d, Supplementary Fig. 2a, b, and Supplementary Table 1). Finally, to assess chromosome arm-specific telomere length, individual reads were anchored to chromosome arms using the subtelomeric sequences (Supplementary Figs. 1c, 2c–e). The normalized coverage of mapped telomeric reads per chromosome arm revealed consistent and uniform coverage across replicates (Supplementary Fig. 2e). The intrasample telomere lengths between different chromosome arms were heterogenous, ranging in the two replicates from median telomere lengths of 3088 and 3139 bp at

chromosome 17q to 9603 and 9835 bp at chromosome 21p (Fig. 1e and Supplementary Data 1). However, despite the intrasample heterogeneity, the chromosome arm-specific telomere lengths were highly reproducible in the two replicates (Supplementary Fig. 2f). We, therefore, conclude that Telo-seq can reproducibly measure bulk and chromosome arm-specific telomere lengths of human cells.

### Telo-seq resolves telomere shortening

In the absence of an active TMM, telomeres progressively shorten due to the end-replication problem and subsequent processing[1,9,10]. To address whether Telo-seq can resolve telomere shortening dynamics, IMR90 human lung fibroblasts expressing the human papillomavirus E6 and E7 oncogenes (IMR90 [E6E7]) were grown in vitro to replicative crisis and sampled at different population doublings (PD) for telomere length analysis[16,17]. In line with the absence of a TMM, TRF analysis revealed that IMR90 [E6E7] telomeres progressively shorten with increasing PDs (Fig. 2a). Similarly, telomere shortening was also detected by Telo-seq (Fig. 2b–d, Supplementary Fig. 3a, and Supplementary Table 2). The IMR90 [E6E7] mean bulk telomere length shortened from 4276 bp at PD66.2 to 2746 bp at PD106.1. The fraction of short telomeres below 1 kb in length was progressively increasing from 5.0% at PD66.2 to 18.3% at PD106.1, while the fraction of telomeres above 10 kb decreased from 3.2 to 1.0% (Fig. 2d and Supplementary Table 2). By plotting the mean telomere length against the PDs and performing linear regression analysis (Fig. 2e), we estimated that IMR90 [E6E7] telomeres shorten on average around 39 bp per PD under the growth conditions used in our laboratory.

Next, we analyzed IMR90 [E6E7] chromosome arm-specific telomere length dynamics (Fig. 2f, Supplementary Fig. 3b, c, and Supplementary Data 2). Similar to the bulk telomere length analysis, IMR90 [E6E7] telomeres of individual chromosome arms progressively shortened with increasing PDs, but at variable shortening rates (Fig. 2f and Supplementary Fig. 3c). As indicated by the HG002 chromosome arm-specific telomere length analysis (Fig. 1e), telomere lengths between different chromosome arms in IMR90 [E6E7] were highly heterogenous (Fig. 2f). For example, at PD66.2 the median telomere length of chromosome 18q was 9182 bp, whereas the median telomere at chromosome 19q was 2262 bp long. Taken together, Telo-seq can measure bulk and chromosome arm-specific telomere length dynamics and resolve telomere shortening rates of samples only five PDs apart.

### Telomere length differs between alleles

Our analysis suggested that bulk telomere length heterogeneity is partly a consequence of the chromosome arm-specific telomere length heterogeneity. We speculated that, similar to the bulk telomere length, the intrachromosomal arm-specific telomere length heterogeneity could partially originate from the two alleles of each chromosome arm. To test this, we first used HG002 to resolve allele-specific telomere length by mapping the reads to the phased HG002 reference genome (Fig. 3, Supplementary Fig. 4a, and Supplementary Data 3). Indeed, some of the intrachromosome arm-specific heterogeneity could be explained by differences in the allele-specific telomere length. For example, the HG002 chromosome arm 1p maternal allele had a median telomere length of 4165 bp, whereas the median paternal allele was 11,139 bp long (Figs. 1e, 3 and Supplementary Data 1, 3). As the mapping approach requires a phased reference genome for a given sample, we tested whether allele-specific telomere length can also be quantified by de novo haplotype phasing (Supplementary Fig. 4b). Indeed, both approaches retrieved comparable allele-specific telomere length information for HG002. However, the strict implementation of phasing introduced certain limitations, resulting in the absence of alignments to some chromosomes and alleles. This can be attributed to two key factors: firstly, conventional phasing algorithms are designed solely for primary alignments and do not consider secondary

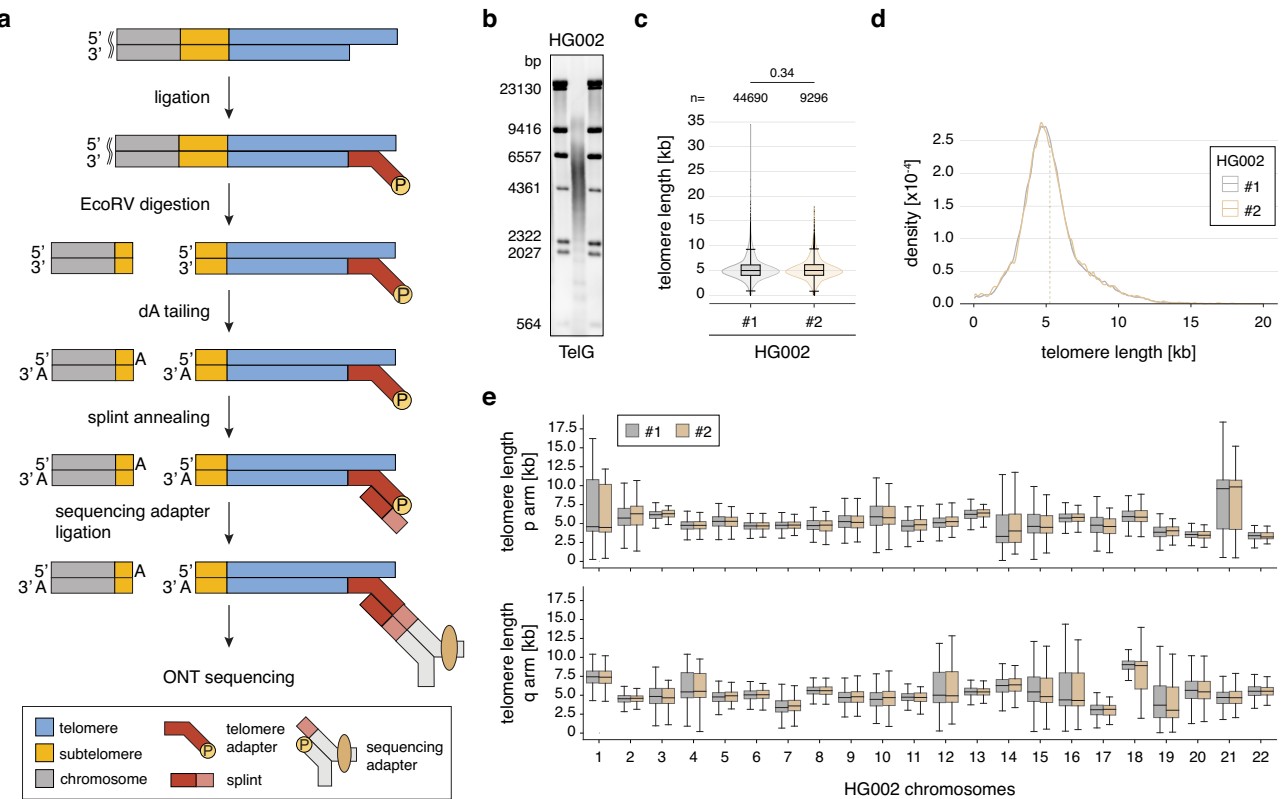

**Fig. 1 | Telo-seq measures bulk and chromosome arm-specific telomere length.**
**a** Schematic overview of Telo-seq protocol. **b** Terminal restriction fragment (TRF) analysis of HG002. **c** Violin plot of HG002 Telo-seq telomere length measurements. Violin represents telomere length distribution in kilobases (kb). Boxplot shows the median telomere length with interquartile range (IQR) and whiskers represent 1.5-fold IQR. The number of telomeric reads per sample is shown above the plot. Statistical analysis two-sided *t*-test with Bonferroni correction was performed and the adjusted *p* value is shown. **d** HG002 Telo-seq telomere length distribution with the mean telomere length shown as the dotted line. **e** Boxplot of the chromosome arm-specific telomere lengths of both HG002 replicates. The middle line represents the median telomere length, box the IQR, and whiskers 1.5-fold IQR. For (**c–e**), the results of two independent HG002 replicates are shown. For limitations on Telo-seq chromosome arm-specific telomere length assignment, see "Discussion". For more details, see Supplementary Table 1, Supplementary Data 1 and Supplementary Figs. 1, 2. TRF: one experiment; Telo-seq: two experiments. Source data are provided as a Source Data file.

alignments. Secondly, using a custom pangenome reference leads to higher quality alignments, compared to solely relying on a single reference, especially when dealing with a diverse population where an exact reference is not available. We applied the de novo haplotype phasing approach to IMR90 [E6E7] PD66.2 in order to resolve IMR90 [E6E7] allele-specific telomere length (Supplementary Fig. 4c). Similar to HG002, the IMR90 [E6E7] allele-specific telomere length was frequently more homogenous than at the chromosome arm-level (Supplementary Fig. 4c). Thus, in addition to bulk and chromosome arm-specific telomere length assessment, Telo-seq allows the analysis of higher resolution allele-specific telomere length.

### Telo-seq on donor-derived fibroblasts revealed shorter telomeres with human age

To investigate telomere length in aged human individuals, we next performed Telo-seq on patient-derived fibroblasts obtained from donors between 20 to 94 years of age. Telomeres were generally longer in the younger individuals than in the older individuals, except for an 82-year-old individual who harbored telomeres of comparable length to the young individuals (Fig. 4a, b, Supplementary Fig. 5a, and Supplementary Table 3). In line with that, the fraction of short telomeres below 1 kb and long telomeres above 10 kb in length were lowest and highest in young individuals, respectively (Supplementary Table 3). Plotting of the mean telomere length against the donor age and linear regression analysis revealed a general trend of bulk telomere shortening as a function of donor age (Fig. 4c). Telomere length assessment of individual

chromosome arms confirmed the intrasample heterogeneity of chromosome arm-specific telomere length observed in HG002 and IMR90 [E6E7] (Supplementary Fig. 5b, c and Supplementary Data 4). Thus, we asked whether any chromosome arms consistently possessed shorter or longer telomeres than the average telomere length in the sample. To address this, we combined the eight fibroblast samples with the IMR90 [E6E7] PD66.2 sample and ranked chromosome arms according to their mean telomere length. Based on these nine individuals and despite the heterogeneity, some chromosome arms consistently had shorter or longer telomeres, relative to the mean telomere length (Supplementary Fig. 5d), raising the possibility that conserved chromosome arm-specific features influence telomere length.

### Telomere length is reset in donor-matched iPSCs

Upon induction of pluripotency, aging hallmarks[41] are reversed, including telomere shortening, which is counteracted by telomerase[42–45]. To investigate the effect of induced pluripotency on telomere length, we performed Telo-seq on six matched induced pluripotent stem cells (iPSCs) of fibroblast donors between 25 and 94 years of age (Fig. 4d, Supplementary Fig. 6a, b, and Supplementary Table 4). Telomeres in iPSCs were, on average, 3202 bp longer than in their matched primary fibroblasts. Furthermore, the fraction of telomeres longer than 10 kb was increased 2- to 12-fold in iPSCs relative to their matched fibroblast controls (Fig. 4e). The dispersion in the telomere length distribution was reduced in iPSCs suggesting that telomerase activity results in more homogenous telomere length

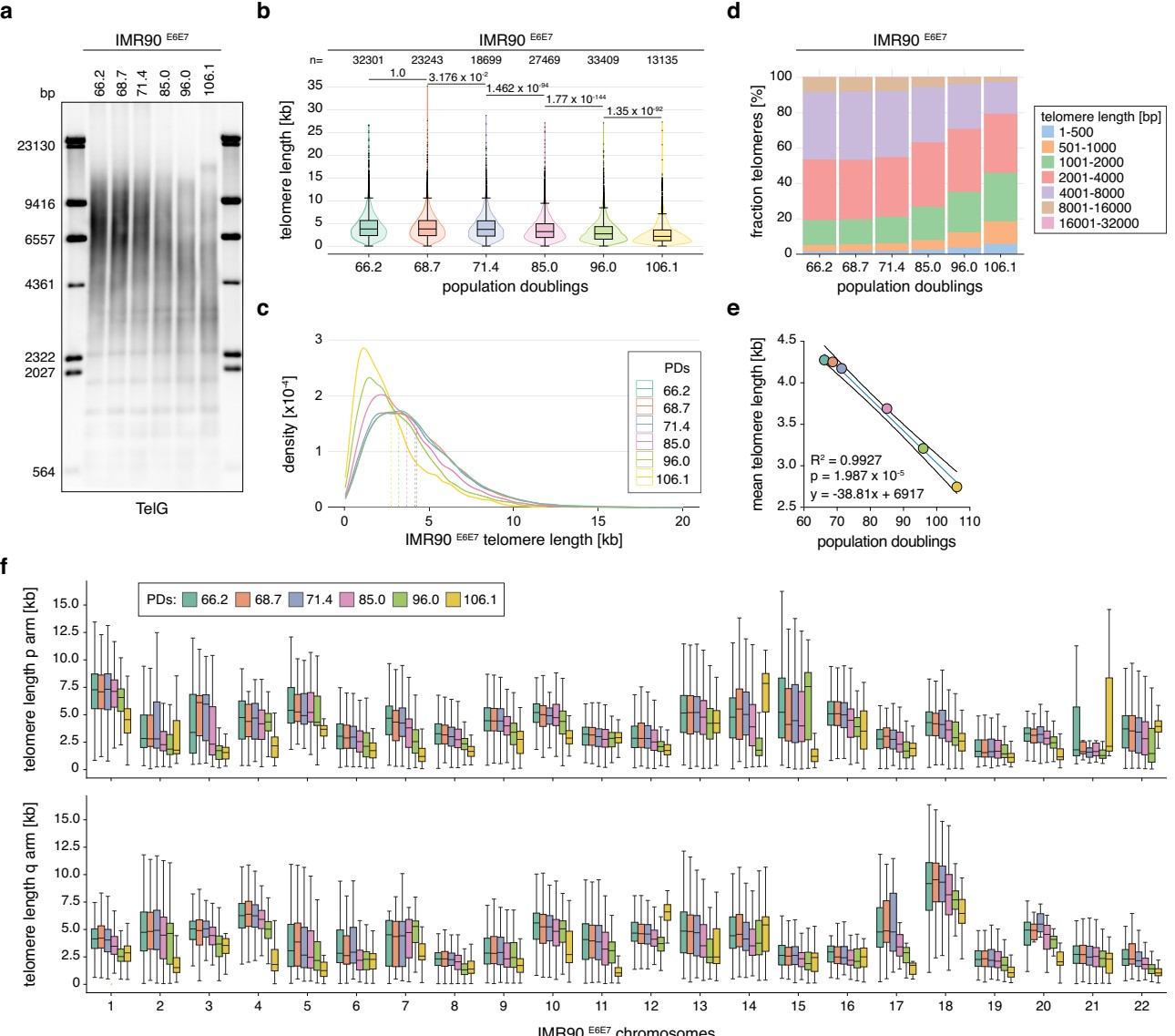

**Fig. 2 | Telo-seq resolves telomere shortening. a** Terminal restriction fragment (TRF) analysis of human IMR90 E6E7 fibroblasts at different population doublings (PDs). **b** Violin plot of IMR90 E6E7 Telo-seq telomere length measurements in kilobases (kb) at indicated PDs. Violin represents telomere length distribution. Boxplot shows the median telomere length with interquartile range (IQR) and whiskers represent 1.5-fold IQR. The number of telomeric reads per sample is shown above the plot. Statistical analysis two-sided *t*-test with Bonferroni correction was performed and adjusted *p* values are shown. **c** IMR90 E6E7 Telo-seq telomere length distribution at different PDs with the mean telomere length shown as the dotted line. **d** Bar graph showing the percentage of binned telomere length in IMR90 E6E7 fibroblasts at different PDs. **e** Linear regression analysis of IMR90 E6E7 mean telomere length against PDs. The blue line represents the best fit and the black curves have 95% confidence intervals. **f** Boxplot of chromosome arm-specific IMR90 E6E7 telomere length at different PDs. The middle line represents the median telomere length, box the IQR, and whiskers 1.5-fold IQR. For limitations on Telo-seq chromosome arm-specific telomere length assignment, see "Discussion". For more details, see Supplementary Table 2, Supplementary Data 2 and Supplementary Fig. 3. One experiment per sample. Source data are provided as a Source Data file.

(Supplementary Tables 3, 4). Independently of the telomere length in matched fibroblasts and the donor's age, iPSCs mean telomeres were set to 8 to 10 kb (Fig. 4d and Supplementary Tables 3, 4), indicating that in this range, human iPSCs' telomeres are in an equilibrium, regulated by telomerase-dependent elongation and telomere trimming[46].

Based on linear regression analysis of chromosome arm-specific mean telomere lengths with at least a 20x coverage in matched fibroblasts and iPSCs (Supplementary Figs. 5b, c, 6c–f and Supplementary Data 4, 5), we concluded that upon induced pluripotency shorter telomeres are preferentially elongated by telomerase; however, the overall order of chromosome arm-specific telomere length remained conserved between fibroblasts and iPSCs.

## Telo-seq distinguishes between TERT+ and ALT+ cancer cells

Cancer cells maintain their telomeres by either reactivation of telomerase or ALT[19–21]. To address the impact of both TMMs on telomeres, we performed Telo-seq on a set of five TERT+ and five ALT+ cancer cell lines. TRF analysis revealed that bulk telomere length in TERT+ cell lines was more tightly distributed, whereas consistent with previous literature[47], the telomeres of ALT+ cancer cell lines were more heterogenous in length (Fig. 5a). Telo-seq analysis recapitulated this striking difference in the telomere length distribution between TERT+ and ALT+ cancer cell lines (Fig. 5b, c and Supplementary Table 5) and was highly reproducible between independent cancer cell line replicates (Supplementary Table 6). In ALT+ cells, we could measure telomeres from 49 bp to 134.7 kb in length (Supplementary Table 5),

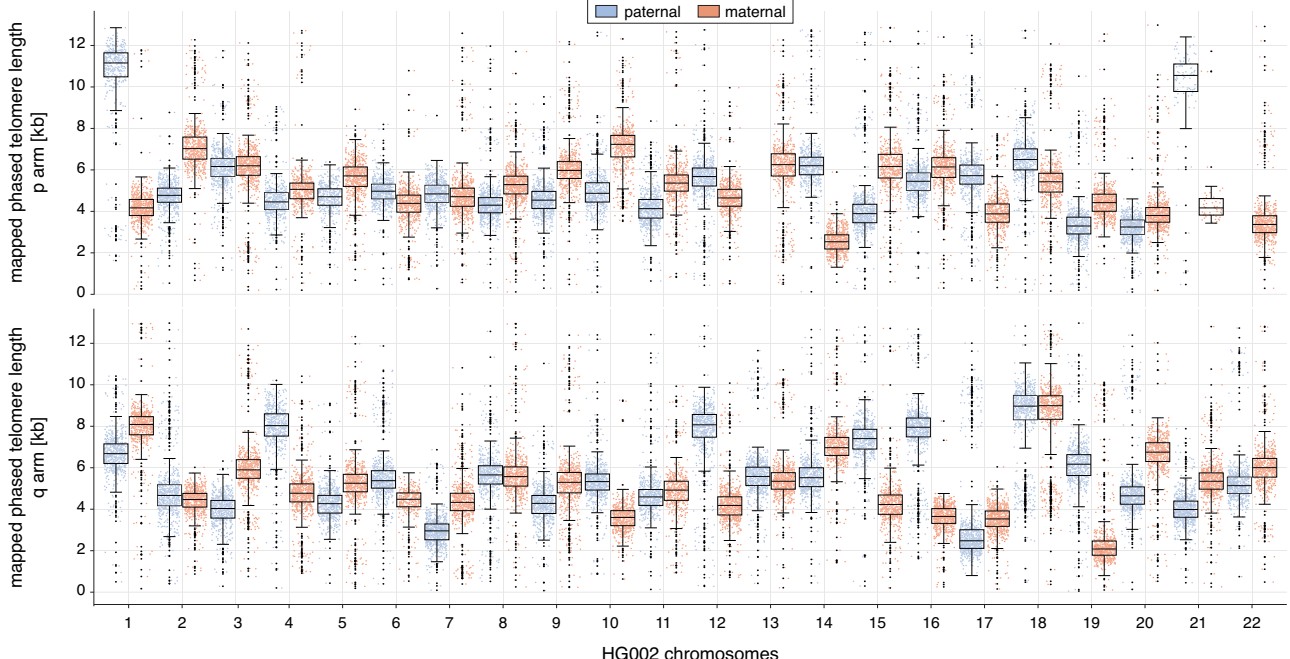

**Fig. 3 | Allele-specific telomere length differs in HG002.** Boxplot of allele-specific telomere length in kilobase (kb) of HG002 based on mapping against the haplotyped HG002 reference genome. The middle line represents the median, box the interquartile range (IQR), and whiskers the 1.5-fold IQR with outliers shown as black points. Individual telomere reads are shown as blue (paternal) and red (maternal) points. The paternal alleles of chromosome 13p and 22p are missing due to not passing the mapping quality filter. For more details, see Supplementary Data 3, Supplementary Fig. 4a and Methods. Pooled data from two independent experiments. Source data are provided as a Source Data file.

suggesting that Telo-seq is an efficient approach to resolve the very long telomeres present in ALT[+] cells. Next, we speculated whether Telo-seq can distinguish between ALT[+] and TERT[+] cancer cells by plotting the coefficient of variation (CV), a measure of the dispersion in a distribution, against mean telomere length (Fig. 5d). Indeed, all ALT[+] cancer cells had a CV above 0.8, whereas the TERT[+] cancer cells showed a CV smaller than 0.55, indicating that Telo-seq is an effective method to postulate TMM of cancer cells.

The subtelomeres of most human chromosome arms contain CpG islands adjacent to the telomere. As Telo-seq uses native DNA, we retrieved the subtelomeric DNA methylation information from the long nanopore reads. In line with previous studies[48], the subtelomeric CpG islands were frequently hypomethylated in ALT[+] cancer cell lines (Supplementary Figs. 7–9). Not all the chromosome arms were equally hypomethylated, and we found differences within and between ALT[+] cancer cell lines. Thus, Telo-seq allows not only telomere length measurements, but also chromosome arm-specific subtelomeric methylation analysis.

## Discussion

Here, we developed Telo-seq, an efficient and reproducible method to determine the length and sequence of entire human telomeres and part of the adjacent subtelomere using Oxford Nanopore Technologies long-read sequencing (Fig. 1a). We demonstrate that Telo-seq resolves the short telomeres present in crisis cells[49] to the very long telomeres existing in ALT[+] cancer cell lines[21]. We find that intrasample telomere length is heterogenous and short telomeres are identified even in cells many population doublings away from proliferation barriers (Fig. 2 and Supplementary Table 2). Harvesting subtelomeric information enabled us to assess chromosome arm and, in specific cases, allele-specific telomere length (Figs. 1e, 2f, 3, Supplementary Figs. 4b, c, 5c, 6d, and Supplementary Data 1–5). Whereas the accuracy of the mean or median telomere length is dependent on the number of underlying reads (Supplementary Fig. 1h), the correct annotation of

reads to specific chromosome arms or alleles depends on at least three additional factors: the subtelomeric sequence length, the subtelomeres homology to other subtelomeres and the sample's similarity to the reference genome. First, a shorter subtelomeric track in the read results in weaker mapping quality and potentially lower chromosome arm coverage. For example, due to an EcoRV cut site close to the subtelomere-telomere boundary, chromosome 5p has the shortest subtelomere length (Supplementary Fig. 2b). For HG002 with a T2T reference available, the 5p coverage is still comparable to other chromosome arms (Supplementary Fig. 2e), but the 5p mapping quality is second lowest of all HG002 chromosome arms (Supplementary Fig. 2d). In other samples without a matching T2T reference genome available, like IMR90 [E6E7] or the fibroblasts of the aging cohort, chromosome 5p coverage is frequently low, likely a consequence of the short subtelomeric track (Supplementary Figs. 3b, 5b). Second, some subtelomeres are similar to one another, which may result in misaligning. For example, the p arms of the human acrocentric chromosomes 13, 14, 15, 21, and 22 were shown to recombine frequently due to their homology[50]. In line with these results, our simulation of HG002 reads suggests that a fraction of these reads misalign between these acrocentric chromosome arms (Supplementary Fig. 2c). Deviation from the expected normalized chromosome arm coverage is also an indicator of potential misalignment. Third, the mapping depends on the sample's similarity to the reference genome. Therefore, to assess chromosome arm-specific telomere length with Telo-seq we apply a pangenome reference consisting of three T2T genomes (Supplementary Fig. 1c). This improves mapping relative to a single genome, however the number of genomes in our pangenome reference is still limited due to the low number of existing T2T genome assemblies. In the future, the number of T2T genomes will increase and represent a more diverse population of human individuals[51]. Including these future T2T assemblies will further improve mapping accuracies. The observation that telomeric reads of TERT[+] cancer cell lines overall map better to the pangenome reference than the frequently genetically

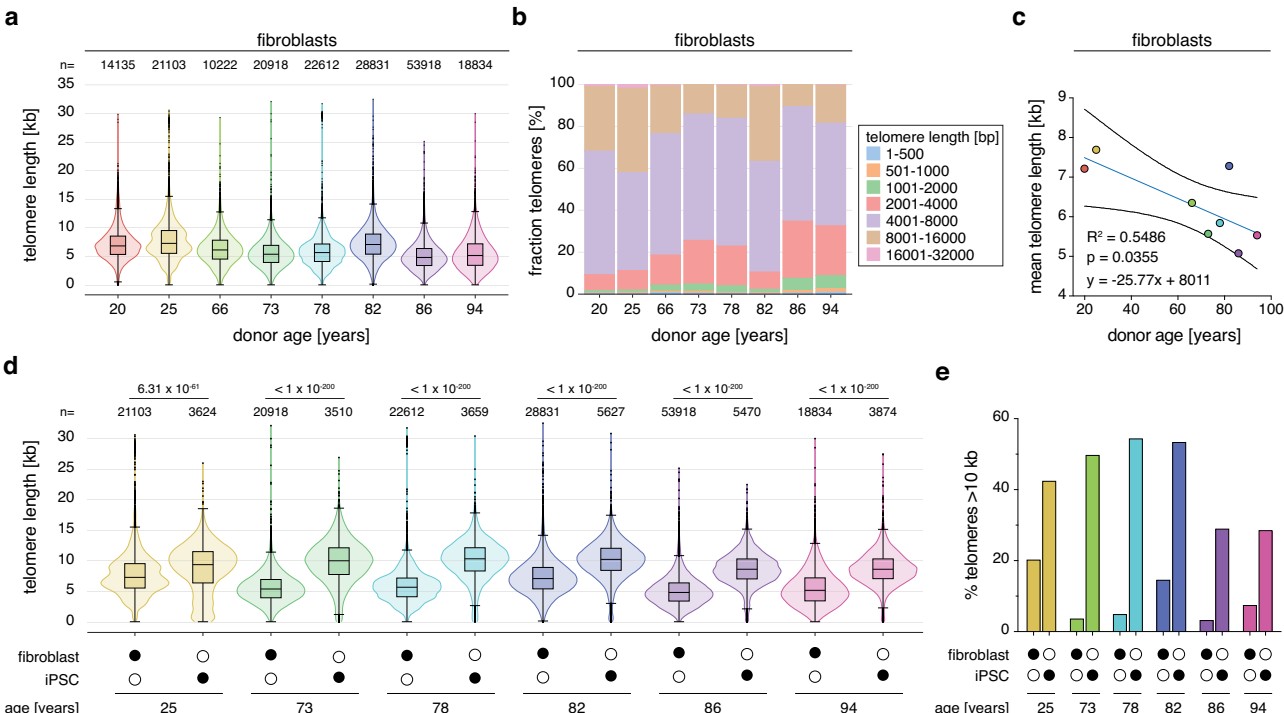

**Fig. 4 | Telomeres shorten with age. a** Violin plots of donor-derived fibroblast Telo-seq telomere length measurements in kilobases (kb). Violin represents telomere length distribution. Boxplot shows the median telomere length with interquartile range (IQR) and whiskers represent 1.5-fold IQR. The number of telomeric reads per sample is shown above the plot. **b** Bar graph showing the percentage of binned telomere length in donor-derived fibroblasts. **c** Linear regression analysis of fibroblast mean telomere length against donor age. The blue line represents the best fit and the black curves have 95% confidence intervals. **d** Violin plots of donor-derived fibroblast and matched induced pluripotent stem cells (iPSC) Telo-seq

telomere length measurements. Violin represents telomere length distribution. Boxplot shows the median telomere length with IQR and whiskers represent 1.5-fold IQR. The number of telomeric reads per sample is shown above the plot. Statistical analysis two-sided *t*-test with Bonferroni correction was performed and adjusted *p* values are shown. **e** Bar graph showing the percentage of telomeres longer than 10 kb of donor-derived fibroblasts and their matched iPSC. For more details, see Supplementary Tables 3, 4, Supplementary Data 4, 5 and Supplementary Figs. 5, 6. One experiment per sample. Source data are provided as a Source Data file.

more unstable ALT+ cancer samples[52] emphasizes the connection between mapping quality and the sample's similarity to the reference genome (Supplementary Tables 5, 6). Thus, Telo-seq is able to resolve chromosome arm-specific telomere length, and, in some cases, allele-specific telomere length; however, the above outlined current limitations should be taken into consideration for the interpretation of the results.

In line with previous work quantifying chromosome arm-specific telomeric FISH staining on metaphases[53–56] and nanopore sequencing of budding yeast telomeres[31], our work shows that chromosome arms, and even alleles, vary in their telomere length distributions within a single sample. Hence, some of the detected bulk and chromosome arm-specific length heterogeneity can be explained by the chromosome arm and allele-specific telomere length variability, respectively.

Furthermore, similar to previous work[54,56] our analysis suggests that some of these chromosome arm-specific telomere length differences are conserved between different individuals (Supplementary Fig. 5d). Nine out of the ten chromosome arms with the shortest and longest telomeres in our cohort were previously reported to be shorter and longer, respectively, than the mean telomere length in a study measuring chromosome arm-specific telomere length with Q-FISH on metaphases of ten individuals[54]. This may indicate that, similar to budding yeast[31], there are conserved chromosome arm-specific factors that influence telomere length homeostasis in humans that are yet to be identified. However, given the telomere length heterogeneity between individuals at the chromosome arm-level and the small cohort sizes, the ranks of specific chromosome arms differ between studies[54] (Supplementary Fig. 5d). Thus, to comprehensively address these questions, a large, diverse and well-controlled human cohort is

required. Telo-seq analysis of this cohort will help to better define and characterize these potentially conserved chromosome arms and, in combination with genome editing and high-content imaging, will serve to identify mechanisms as to why these chromosome arms consistently differ from the average telomere length.

Telomere shortening is a hallmark of aging[41] and an elegant protective mechanism to prevent infinite proliferation[17]. In line with previous telomere shortening rates for human fibroblasts[9,57], we find that the mean telomere length in IMR90 E6E7 fibroblasts shorten by ~39 bp per PD (Fig. 2e). Interestingly, individual chromosome arms vary in their shortening rate (Supplementary Fig. 3c), likely again pointing at chromosome arm-specific factors that influence telomeric processing[1]. Similarly, telomere length and attrition may vary in different tissues and cell types of the same individual depending on tissue-specific differential gene expression, the cellular turnover rate, the exposure to endogenous and exogenous sources of DNA damage and other stress factors, as well as inflammation[58]. Telo-seq on tissues or sorted cell types will not only reveal precise telomere length measurements, but also provide information about telomeric sequence composition and subtelomeric methylation status during human aging and disease.

Consistent with previous reports[43–45], we show that upon induction of pluripotency, the mean telomere length is reset independently of the parental cell's initial telomere length (Fig. 4d). Our results indicate that telomerase preferentially elongates shorter telomeres (Supplementary Supplementary Fig. 6e), potentially a result of different shelterin accessibility, coverage or stoichiometry, or altered chromatin structure at short telomeres compared to long ones. As a result, this not only leads to increased bulk telomere length (Fig. 4d and

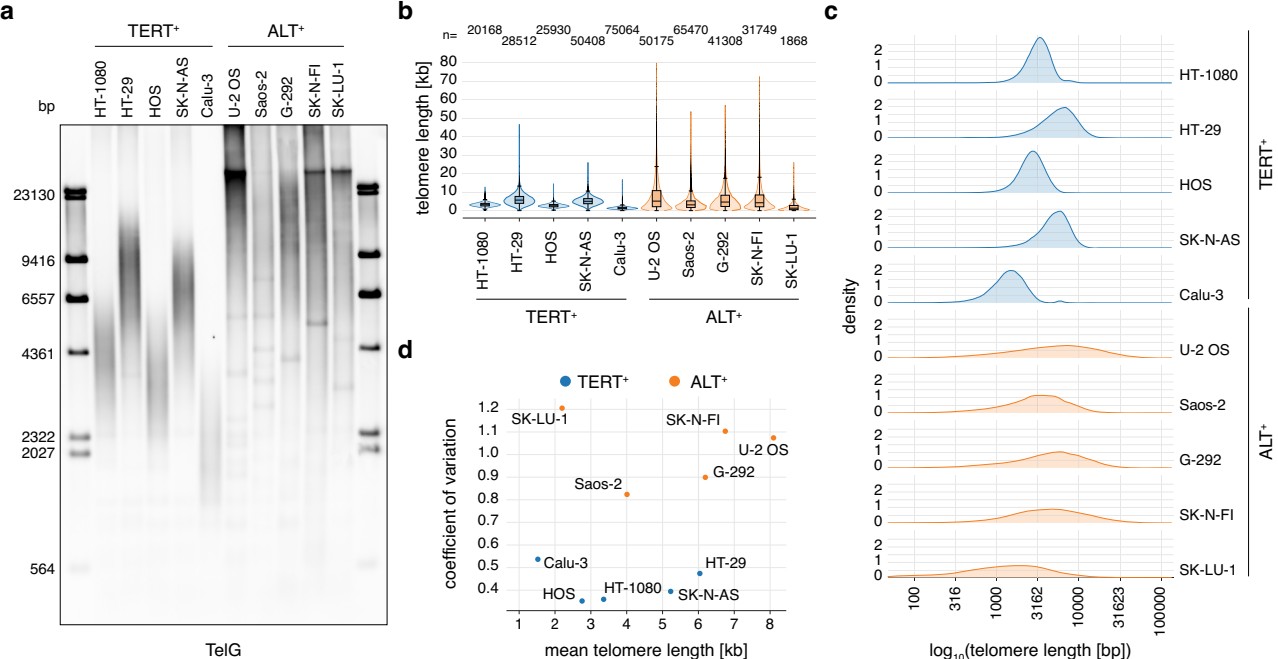

**Fig. 5 | Telo-seq resolves TERT+ and ALT+ cancer cell telomere length distributions. a** Terminal restriction fragment (TRF) analysis of the indicated cancer cell lines. **b** Violin plot of cancer cell line Telo-seq telomere length measurements. Telomeres shorter than 80 kb are shown. Violin represents telomere length distribution. Boxplot shows the median telomere length with interquartile range (IQR) and whiskers represent 1.5-fold IQR. The number of telomeric reads per sample is shown above the plot. **c** Density plot of telomere length distributions measured by Telo-seq in indicated cancer cell lines. Telomere length in base pairs (bp) is log10 transformed. **d** Scatter plot of coefficient of variations to mean telomere length in kb of indicated cancer cell lines. For more details, see Supplementary Tables 5, 6. TRF: one experiment; Telo-seq: one, two, or four experiments. See Supplementary Table 6 for more information. Source data are provided as a Source Data file.

Supplementary Fig. 6b), but also a more homogenous telomere length distribution (Supplementary Tables 3, 4).

Moreover, our analysis of cancer cells reveals that Telo-seq can efficiently distinguish between TERT+ and ALT+ cancer cells (Fig. 5b–d), establishing Telo-seq as a reliable method of TMM prediction, an essential prerequisite to target TMM-specific vulnerabilities in a personalized cancer therapy[22]. Further, in combination with whole-genome sequencing, Telo-seq will help to investigate inherited telomere syndromes and resolve the impact of the underlying genetic alterations on telomere structure[59]. In summary, our study highlights the potential of human telomere long-read sequencing and sets the stage to investigate human telomere dynamics in unprecedented detail during development, aging and disease.

## Methods
Our research complies with all ethical regulations as dictated by the Salk Institute's Code of Conduct, IACUC, Biosafety and IRB committees, and the Academic Council.

### Cell culture
Calu-3 (HTB-55), G-292, clone A141B1 (CRL-1423), HOS (CRL-1543), HT-1080 (CCL-121), HT-29 (HTB-38), Saos-2 (HTB85), SK-LU-1 (HTB-57), SK-N-AS (CRL-2137), SK-N-FI (CRL-2142), and U-2 OS (HTB96) cancer cell lines and IMR90 (CCL-186) fibroblasts were purchased from ATCC. The lymphoblastoid cell line HG002 (GM24385) was purchased from Coriell Institute for Medical Research. Donor-derived fibroblasts were collected at the University of California, San Diego (UCSD) and are part of the Salk AHA-Allen aging cohort. Alzheimer's Disease Research Center participants at UCSD have given broad consent to a range of experiments, including skin fibroblast and induced pluripotent stem cell derivation, cell engineering, and genetic sequencing and manipulation prior to providing a skin biopsy.

All cells were grown at 37 °C under 7.5% $CO_2$ and 3% $O_2$, except HG002, and induced pluripotent stem cells (iPSCs) were grown at 37 °C under 5% $CO_2$ and ambient $O_2$. IMR90 fibroblasts were grown in GlutaMax-DMEM (Gibco, 10569-010) supplemented with 0.1 mM non-essential amino acids (Corning, 25-025-Cl), 15% fetal bovine serum (FBS) (VWR, 97068-085 Lot 323B20), 100 IU/mL Penicillin and 100 µg/ml Streptomycin (Corning, 30-002-CI). Donor-derived fibroblasts were grown in GlutaMax-DMEM supplemented with 0.1 mM non-essential amino acids, 20% FBS, 20 ng/mL FGF-2 (Joint Protein Central), 100 IU/mL Penicillin, and 100 µg/ml Streptomycin. Calu-3, HOS, HT-1080, Saos-2, SK-LU-1, SK-N-AS, SK-N-FI, and U-2 OS were grown in GlutaMax-DMEM supplemented with 0.1 mM non-essential amino acids, 10% FBS, 100 IU/mL Penicillin and 100 µg/ml Streptomycin. HT-29 and G-292 were grown in McCoy-s 5a (modified) media (Gibco, 16600-108) supplemented with 10% FBS and 100 IU/mL Penicillin and 100 µg/ml Streptomycin. HG002 cells were grown in Roswell Park Memorial Institute 1640 medium (ATCC, 30-2001) and 10% FBS (ATCC, 30-2020) and 1% antibiotic antimycotic (Gibco, 15240062). iPSCs were grown in StemMACS™ iPS-Brew XF, human (Miltenyi, 130-104-368) with daily medium replacement.

The number of population doublings (PD) of IMR90 fibroblasts were calculated using the following equation: PD = log(cells harvested/cells seeded)/log2. Cells have been tested to be free of mycoplasma.

### Induced pluripotent stem cell generation
Induced pluripotent stem cells (iPSCs) were generated from donor-derived dermal fibroblasts at the Salk stem cell core according to standards of the International Society for Stem Cell Research standards[60]. In brief, dermal fibroblasts were expanded and verified mycoplasm negative via MycoAlert™ PLUS Mycoplasma Detection Kit (Lonza, 75860-362) infected with Sendai virus containing Yamanaka factors from the CytoTune™-iPS 2.0 Sendai Reprogramming Kit (Thermo Fisher, A165167) according to manufacturer

recommendations. iPSCs were grown in medium-sized colonies on Matrigel (BD Biosciences, 354230) at a final concentration of 1 mg per six-well plate and propagated in StemMACS™ iPS-Brew XF, human (Miltenyi, 130-104-368) with daily medium replacement.

## Terminal restriction fragment analysis

Terminal restriction fragment (TRF) was performed as previously described[15,61]. In brief, high-molecular-weight genomic DNA (gDNA) was isolated by phenol-chloroform extraction and digested with 50 U AluI (NEB, R0137L) and 50 U MboI (NEB, R0147M) at 37 °C, overnight. Digested gDNA was quantified using Qubit dsDNA BR assay (Invitrogen, Q32850), and either 4 or 5 μg digested gDNA was separated on a 0.7% agarose gel overnight in 1x TAE buffer at 40 V. Next, the gel was incubated in depurination buffer (0.25 M HCl) for 10 min followed by two 15 min incubations in denaturing buffer (1.5 M NaCl, 0.5 M NaOH) and two 15 min incubations in neutralization buffer (1 M Tris, pH 7.4, 1.5 M NaCl). The gDNA was transferred to a positively charged nylon membrane (Amersham, RPN203B), overnight. After UV-crosslinking, the membrane was incubated in prehybridization buffer (5x SSC, 0.1% N-lauroylsarcosine sodium salt and 0.04% sodium dodecyl sulfate (SDS)) for 2 h at 65 °C followed by hybridization overnight at 65 °C (1.3 nM digoxigenin-labeled TelG probe prepared according to ref. 62 in prehybridization buffer). The next day, the membrane was washed with wash buffer 1 (2x SSC + 0.1% SDS) for 15 min, thrice, followed by one 15-min wash in wash buffer 2 (2x SSC). The membrane was blocked with freshly prepared blocking solution (100 mM maleic acid, 150 mM NaCl, pH 7.5, 1% (wt/vol) blocking reagent (Roche, 11096176001)) for 30 min. Next, the membrane was incubated with anti-digoxigenin-AP Fab fragments (Roche, 11093274910, 1:2000 dilution in blocking solution) for 30 min. The membrane was washed twice with wash buffer 3 (1x maleic acid buffer, 0.3% Tween 20) for 15 min and equilibrated with AP buffer (100 mM Tris, 100 mM NaCl, pH 9.5) for 2 min. Blot was developed using CDP-star ready-to-use solution (Roche, 12041677001) and imaged in G-box (Syngene). Telomere length were quantified using TeloTool (version 1.3)[63] and WALTER (version 2.0)[64].

## Genomic DNA extraction and purification for Oxford nanopore technologies sequencing

Genomic DNA was extracted from $5-10 \times 10^6$ cells using the Gentra Puregene Cell Kit (Qiagen, cat no. 158046) following the manufacturer's instructions. Extracted DNA was further purified by isopropanol precipitation DNA was quantified using the Qubit fluorometer (Thermo Fisher).

## Whole-genome sequencing

Of the extracted genomic DNA, 4 μg was sheared following the Covaris g-tube (520079) manufacturer's instructions. About 1 μg of the sheared DNA was prepared for sequencing following the Standard Ligation Sequencing Kit instructions (Oxford Nanopore Technologies, catalog no. SQK-LSK110). Libraries were sequenced on a GridION sequencer (Oxford Nanopore Technologies) with R9.4.1 flow cells. One flow cell was used per library.

## AluI/MboI restriction digestion for sequencing

In a final volume of 400 μL, 15 μg extracted genomic DNA was incubated with 20 μL MboI (NEB R0147, 5 U/μL) and 20 μL AluI (NEB R0137, 10 U/μL) restriction enzymes and 0.2 μL RNase A (NEB T3018, 20 mg/mL) in 1x rCutSmart buffer. The digestion reaction was incubated at 37 °C for 16 h and then heat inactivated at 80 °C for 20 min. The resulting DNA was purified using 0.5x v/v AMPure XP (Beckman Coulter, A63881) and eluted into 70 μL of water. Of the digested DNA, 1 μg was taken forward and prepared for sequencing following the Standard Ligation Sequencing Kit instructions (Oxford Nanopore

Technologies, catalog no. SQK-LSK109). Libraries were sequenced on a GridION sequencer (Oxford Nanopore Technologies) with R9.4.1 flow cells. One flow cell was used per library.

## Telo-seq

Six 5'-phosphorylated single-stranded oligo adapters containing permutations of an 18 bp sequence complementary to the human telomeric repeat were combined in equal parts to a final concentration of 1 μM. Oligo sequences are shown in Supplementary Table 7. A separate splint oligonucleotide S1 with sequence complementarity to the Oxford Nanopore AMII sequencing adapter was diluted to a final concentration of 10 μM. The pre-mixed oligos were ligated with T4 DNA ligase (NEB, M0202) using the following reaction conditions[24]: In a final volume of 200 μL, 15 μg extracted DNA and 20 μL pre-mixed oligos were incubated with 10,000 U T4 DNA ligase in 1x rCutSmart Buffer (NEB, B6004S) supplemented with 0.4 mM ATP (NEB, P0756). The ligation mixture was incubated at 35 °C for 16 h and then heat inactivated at 65 °C for 10 min. Adapter-ligated DNA was subsequently digested with 80 U EcoRV-HF (NEB, R3195) at 37 °C for 30 min, and heat inactivated at 65 °C for 20 min. Digested DNA was treated with 45 U Klenow Fragment (3' → 5' exo-) (NEB, E6053) in a final volume of 250 μL of 1x NEBNext dA-tailing reaction buffer (NEB), split evenly into five 50 μL aliquots, and incubated at 37 °C for 30 min. The resulting DNA was purified using 1x v/v AMPure XP (Beckman Coulter, A63881) and eluted into 190 μL of water at 37 °C for 15 min. In the presence of 50 mM NaCl and 100 nM splint oligo, purified DNA was annealed at 50 °C for 1 h. Finally, DNA was purified using 0.5x v/v AMPure XP and eluted into 30 μL of water at 37 °C for 15 min. Beginning at the sequencing adapter ligation step, sequencing libraries were prepared with the AMII sequencing adapter following the Native Barcoding Kit instructions (Oxford Nanopore Technologies, SQK-NBD111). Libraries were sequenced on a GridION sequencer (Oxford Nanopore Technologies) with R9.4.1 flow cells. One flow cell was used per library.

## Telomere model training

The telomere-specific model was trained using Bonito (version 0.6.2) on the telomere-to-telomere CHM13v1.1 [https://s3-us-west-2.amazonaws.com/human-pangenomics/T2T/CHM13/assemblies/chm13.draft_v1.1.fasta.gz] reference and the ultra-long CHM13 dataset from the publicly available "Telomere-to-Telomere" (T2T) consortium [https://github.com/marbl/CHM13/blob/master/Sequencing_data.md]. The training data were filtered for reads greater than 50 KB and a primary alignment mapping quality of 60. Only chromosomes with at least 10x coverage of both telomere and adjacent subtelomeres were used for fine-tuning the R9.4.1 sup model with the following parameters (bonito train --epochs 2 --lr 1e-5). The training set included both C- and G-rich telomeres. To prevent overtraining, the number of training iterations were limited to two rounds. The trained Telomere model is deposited in the GitHub repository (see Code availability).

## Analysis

**Telomere identification and length determination.** Reads were basecalled using a telomere Bonito basecalling model trained for calling telomeric repeats. Identification of telomere-containing reads was achieved through the implementation of the Noise Cancelling Repeat Finder (NCRF) algorithm (version 1.01.00 20190426)[65]. Subsequent reads were further filtered based on three criteria:

1. Due to the nature of Oxford Nanopore sequencing reads from the 5' to 3' direction, reads that did not commence with the 5'-$(CCCTAA)_n$ motif or conclude with the 5'-$(TTAGGG)_n$ motif were excluded from the analysis.

2. Identification of the telomere motif within 200 base pairs of the read termini.

3. The inclusion of 60 bp unique, non-telomeric (subtelomeric) sequence adjacent to the telomere motif to select reads without any breaks in the telomeric track.

Together, these three filters select for reads that contain terminal telomeric sequences and adjacent non-telomeric sequences. This prevents telomeric reads with breaks in the telomeric sequence from contributing to the telomere length measurement and potentially resulting in incorrect telomere length determination.

Telomere length determination was carried out by utilizing the outcomes obtained from NCRF. In cases where a read contained continuous telomeric variations spanning over 50 base pairs, these segments were concatenated into a unified sequence, provided that the breakpoints between segments did not exceed 250 base pairs. This concatenated sequence was then considered a single, uninterrupted length measurement for the telomere.

This approach to telomere length determination accommodates potential insertions or variations within the telomeric track. Moreover, analysis of the CHM13 reference dataset demonstrated that the 250-base pair breakpoint segments adequately corresponded to the telomere lengths observed in the CHM13 reference dataset, confirming the effectiveness of the chosen methodology.

To estimate the number of telomeric reads required for a robust telomere length determination within a certain standard error of the mean, modeling was conducted using a custom Python script. The script randomly subsampled 10 times the indicated number of reads from the bulk telomere length distribution of the HG002 control Telo-seq dataset (Fig. 1c, d and Supplementary Table 1) and characterized the generated distributions (Supplementary Fig. 1h).

### Read mapping and alignment

All filtered reads were mapped against a custom reference containing T2T consortium reference genomes HG002 (v0.7)[34,38] and CHM13 (v2.0)[34]. For HG002, 25 kb on each end of the chromosomes for both maternal and paternal copies were retained; for CHM13, the entire genome was retained. Filtered reads were mapped using Minimap2 (version 2.22), with specific alignment parameters optimized for nanopore reads (minimap -x map-ont -N 2 -Y -y -L -a). Alignments were ranked following a hierarchical order of primary, secondary, and supplementary alignments. In scenarios where the primary alignment lacked intersection with both subtelomeric or telomeric regions, secondary alignments were examined iteratively until the intersection criteria were fulfilled. In cases where secondary alignments met the above criteria, they were promoted to primary status for compatibility with downstream analyses, and the original primary alignments were subsequently reclassified as secondary. For chromosome arm-specific telomere length assignment, only telomeric reads with a mapping quality of 20 or higher were used, except for the cancer samples for which telomeric reads with a mapping quality of 10 or higher were included.

### Normalized coverage

To evaluate the bias of chromosomes during alignment, the following steps were implemented. First, the data were filtered for reads that mapped to the telomere and subtelomere regions with a mapping quality of 20 or higher. Second, the coverage per chromosome arm was normalized by the total number of mapped telomeric reads with a mapping quality equal to or greater than 20.

### Simulations

The homology and mapping efficacy of chromosomes was assessed in a simulation study. To accomplish this, reads derived from CHM13 (v2.0)[34] and HG002 (v0.7)[38] were utilized and incorporated a simulated averaged 2% error rate. The methodology involved randomly trimming the telomeres within the range of 0 to 3000 base pairs in both the P

and Q arms of the chromosomes. Each chromosome end was subjected to this process 30 times to simulate 30x coverage per chromosome arm and ensure robustness in the analysis. Subsequently, a virtual digestion of the simulated reads at EcoRV sites was performed and simulated reads were mapped back to the corresponding reference genomes. This mapping process was facilitated using minimap2 with specific parameters (minimap -x map-ont -N 2 -Y -y -L -a). Following mapping, primary alignments were retained and a confusion matrix was generated to evaluate the accuracy of the mapping (Supplementary Fig. 2c).

### Haplotyping

Fully phased reads were obtained by mapping telomere reads from HG002 to the phased diploid reference HG002(v0.7)[34] using Minimap2. Maternal and paternal reads were obtained based on their primary alignment to the contig, filtering for reads with a mapping quality greater than 5 (Fig. 3 and Supplementary Fig. 4a).

For samples where a phased reference genome is not available, a de novo haplotyping approach to quantify allele-specific telomere lengths was developed (Supplementary Fig. 4b, c). Telomere reads were aligned using Minimap2 to the paternal haploid copy of the reference genome HG002 (v0.7)[34] and variants were called using Clair3 with the r941_hac_g360_g422_1235 model. The resulting reads were haplotagged with WhatsHap haplotag.

Haplotagged reads were cross-referenced with the prior alignments obtained using the custom pangenome reference and only reads that matched precisely on read id, chromosome, arm, and strand were considered (see Supplementary Fig. 4b, c). Phased reads with primary alignments to the haploid reference are shown; secondary alignments were not considered.

### Methylation

Telomeric reads were basecalled with Dorado (version 0.3.4) using the dna_r9.4.1_e8_sup@v3.3_5mC_5hmC model. Subsequently, the reads were aligned to a custom pangenome reference using Minimap2. To align methylation sequences with telomere sequences, reads that intersected with subtelomere regions from Telo-seq alignments were retained, taking into account soft-clipping of telomeres. It is important to note that the methylation (dna_r9.4.1_e8_sup@v3.3_5mC_5hm) model was not specifically trained for telomeres. Methylation calls were obtained using Modkit (v0.11.1) pileup with the "traditional" preset. A minimum coverage threshold of ten reads was imposed (Supplementary Figs. 8, 9). For comparative analysis across samples, a heatmap was generated by computing the median methylation values. This computation was conducted while ensuring a minimum coverage threshold of ten reads and a minimum of 100 CpG sites (Supplementary Fig. 7).

### Statistical analysis

The statistical analysis was conducted using GraphPad Prism (version 8.4.3), R (version 4.3.1), and Python (version 3.8.10). In Python, the telomere length distributions were compared using the two-sided $t$-test of independence with Bonferroni correction. For ranking chromosomes, the z-score was calculated with the Python library Scipy.

### Reporting summary

Further information on research design is available in the Nature Portfolio Reporting Summary linked to this article.

## Data availability

The telomere sequencing data generated in this study have been deposited in the BioProject database under accession code PRJNA1040425. Source data are provided as a Source Data file. Source data are provided with this paper.

## Code availability

The code used for telomere length analysis is deposited at https://github.com/priyesh000/teloseq[66].

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

## Acknowledgements

We thank the entire Karlseder lab for the discussions. T.T.S. has received support from the European Molecular Biology Organization (ALTF 668-2019) and the Paul F. Glenn Center for Biology of Aging Research at the Salk Institute. This work was supported by the Stem Cell Core Facility of the Salk Institute with funding from the Helmsley Charitable Trust, the Shiley-Marcos Alzheimer's Disease Research Center (ADRC; AG062429) at the University of California, San Diego (UCSD), the National Institute of Aging (P30AG068635, AG0773424), the National Cancer Institute (CA227934, CA234047, and P30CA014195) and the National Institute for General Medicine (GM142173). F.H.G. received support from the JBP Foundation (grant #2021-2961). This research was supported by an AHA-Allen Initiative in Brain Health and Cognitive Impairment award made jointly through the American Heart Association and The Paul G. Allen Frontiers Group: 19PABH134610000. We thank S. Artandi and S.E. Sanchez for exchanging information and coordinating the manuscripts.

## Author contributions

T.T.S., C.T., P.R., S.H., and J.K. conceived the study. T.T.S., C.T., P.R., X.D., S.H., and J.K. designed experiments. T.T.S., C.T., P.R., C.H., X.D., and S.H conducted experimentation and analyzed data. J.R.J., K.A.F., and F.H.G. provided donor-derived fibroblasts and iPSCs. S.J., S.H., and J.K. supervised the study. T.T.S., C.T., P.R., S.H., and J.K. wrote the manuscript with editorial input from all authors.

## Competing interests

C.H., J.R.J., K.A.F., F.H.G., and J.K. declare that they have no competing interests. T.T.S. has received travel funds to speak at events hosted by Oxford Nanopore Technologies. C.T., P.R., X.D., S.J., and S.H. are employees of Oxford Nanopore Technologies, Inc. and are stock or stock option holders in Oxford Nanopore Technologies plc. C.T., P.R., X.D., and S.H. are named as inventors on a patent application covering aspects of this work filed by Oxford Nanopore Technologies plc. Oxford Nanopore Technologies products are not intended for use for health assessment or to diagnose, treat, mitigate, cure, or prevent any disease or condition.
