## [Peer Review File · Nature Communications]

High resolution long-read telomere sequencing reveals dynamic mechanisms in aging and cancerREVIEWER COMMENTS

Reviewer #1 (Remarks to the Author):

Tobias T et al have demonstrated the capability to measure telomere length using Oxford Nanopore long-read sequencing platform. The authors also design a method (Telo-seq) to enrich for telomere containing genomic DNA, which resulted in more than 75-fold increase in telomeric read output comparing to whole genome sequencing. A customized Bonito telomere model was developed for basecalling and sequencing data analysis, which should be useful for scientists studying telomere length in cancer and aging. Allele-specific telomere length can be measured for all 44 chromosomal ends excluding the sex-determinant chromosomes, showing potential chromosome arm-specific length heterogeneity. In addition to these findings, the authors also apply this new telomere sequencing method to successfully differentiate TERT+ cancer cells and ALT+ cancer cells by measuring the dispersion of their telomere length distribution.

Overall, the results are clear, and the method is interesting. However, the current manuscript is missing key details for comprehensive evaluation of the data, which may limit this new approach to recover accurate mean telomere lengths as well as allele-specific telomere lengths.

Major comments:

1. The Telo-seq method adapted a similar strategy as STELA by ligating telomere-based telomere adapters to the C strand of telomeric ends. This will inadvertently result in loss of information for telomere G tail overhangs, which is known to vary from 100-300 nucleotides in length in human cells. Therefore, the newly established Telo-seq may result in underestimation of actual telomere length. Given the peripheral blood mononuclear cells telomere shortening rate is about 40-50 nts per year, whether this method can be utilized for accurate measurement of telomere length shortening rate in general population remains to be validated.
2. The customized Bonito telomere model that was developed for basecalling in Oxford Nanopore sequencing data analysis will be valuable to the research interest of telomere length in cancer and aging. Please explain the detailed training process for this new model. Please also describe the detailed normalization steps that have been applied to read coverage.
3. Please show the raw read coverage in all figures, which is essential to judge the accuracy of telomere length measurement.
4. The telomere length estimated from HG002 Chr. 21p before haplotype phasing is above 9kb (Fig. 1e and lines 107-108). However, the lengths were estimated to be 4-5kb and 10-11kb for maternal and paternal alleles separately (Extended Data Fig. 4a). So, if the two alleles have been sequenced in similar depth, the telomere length of Chr. 21p should be much smaller than 9kb. Chr.1p has the same problem. Could you elaborate more regarding these differences. In addition, the allele-specific telomere lengths for paternal Chr. 13p and Chr. 22p are missing in Extended Data Fig. 4a as well.
5. In Extended Fig. 4a, it is very clear that both paternal and maternal allele in HG002 chr18q have two sets of reads in it: One is about 4-6kb, the other is about 8-10kb. Why the two alleles in this specific arm exhibit 2 sets of telomeres with different length?
6. It is known that telomere length is reversely correlated with age. However, the authors have identified an 82-year-old individual who has abnormal longer telomere compared to the other individuals in his/her age group (Fig. 3a). How many replicates have been sequenced using Oxford Nanopore platform for each individual? Have the authors compared using the traditional TRF method to measure the telomere length in these eight individuals to validate the accuracy of their telomere length estimation using Oxford Nanopore platform.

7. While Telo-seq may be able to distinguish TERT+ cancer cells and ALT+ cancer cells, the telomere length distribution in ALT cells obtained from Nanopore sequencing seem to be much shorter than the traditional teloblot (compare Fig. 4b and Fig. 4a). What is the potential cause for such discrepancy? The intense high molecular weight fractions in ALT samples seem to indicate incomplete genomic DNA digestion as well.

8. The author indicated that "In ALT+ cells, we could measure telomeres from 47 bp to 134.7 kb in length." The 47 bp telomere seem to be too short. How often does the sample show telomere reads with such short length (percentage)?

It is highly possible that such fragments (47 bp) may be resulted from random shearing of genomic DNA during genomic DNA extraction.

9. The Allele-specific telomere length measurement is certainly one of the most interesting results in this manuscript. While the authors showcase the capability to measure the telomere length of all 88 alleles, whether they are accurate remains a major question.

For example: While the author highlighted their ability to accurately measure telomere shortening rate (average 39 bp per PD) in IMR90E6E7 cells in serial passage in vitro, the allele-specific telomere length shortening seems to have a lot of heterogeneity. In many instances (Fig. 2f, Chr. 22p PD106.1; Chr. 14p PD106.1; Chr 12q PD106.1, etc) the allele-specific telomere lengths in the late passage are much longer than IMR90E6E7 in earlier passages. These discrepancies of allele-specific telomere length bring a lot of doubt on the accuracy of allele-specific telomere length measurement using current method.

In addition, only the chromosomal end specific telomere lengths are presented, but not the allele-specific telomere lengths of IMR90E6E7 cells are shown for each PD? Given the differences of allele-specific telomere length (Extended Data Fig. 4c), plotting the allele-specific telomere shortening rate in Fig. 2f will be important.

10. By profiling allele-specific telomere length using donor-derived fibroblasts, the authors have indicated that some chromosome arms consistently had shorter or longer telomeres. They further proposed that there are potential conserved chromosome arm-specific features influence telomere length. Similar results have been reported previously (Londono-Vallejo J.A. NAR 2001; Martens U.M. Nature Gen. 1998; Graakjaer J. Mech. Ageing Dev. 2003 etc) using other telomere length measurement methods, such as FISH. However, there are no consistency on which chromosomal end has the shortest or the longest telomere.

Similar results were observed in Extended Data Fig. 4a: most of the two alleles from the same chromosomal end seems to have vast different telomere length. The allele-specific telomere length for the eight samples should be shown in Extended data Fig. 3c. In summary, whether there is conserved chromosome arm-specific features influence telomere length remains to be addressed.

11. What is the meaning of the methylation data? There is a lot of heterogeneity, which may be due to the sequencing method itself or read depth. Is the methylation results shown in each panel in extended data Fig. 6 average of multiple reads from the same allele? The results have shown that ALT+ cancer cell lines were observed to be hypomethylated compared to TERT+ cancer cell lines. However, it is also mentioned that not all the arms were equally hypomethylated. Can the authors specifically list out the hypomethylated/hypermethylated chromosome arms? And within the differentially methylated arms, what are the differentially methylated regions (DMR)? Does the ALT+ cell lines have similar DMR compared to TERT+ cell lines? Furthermore, it is also important to compare the methylation pattern of cancer cells with normal cells (HG002 and IMR90).

Minor comments:

1. As indicated by the authors, Telo-seq resulted in "increased telomeric reads 75-fold relative to whole-genome sequencing (Extended Data Fig. 1d)." Such enrichment may pose a problem for high-throughput telomere length application as the author has pointed out that "the telomeric content of human diploid cells is approximately only 0.015% of the total genome." The number of samples that can be applied in single Nanopore flow cell should be mentioned.
2. Are the two replicates of HG002 shown in Fig. 1 obtained in the same or different Nanopore flow cell?
3. In Extended Data Fig. 2c, the shortening rate of 20q, 21p, 1q, and 1p are not significant. Is that because of low sequencing depth of these arms? If so, this analysis can be refined by including more reads.
4. Line 174, the author mentioned that pangenome reference could result in higher alignment quality compared to haploid reference genome. Please give a quantitative measurement of higher quality.
5. Extended Data Figure legends:
Please use either "Extended Data Fig. 1-7" or "Fig. S1-7" to be consistent.
6. Line 259 – Fig 3d missing a ".", Fig. 3d.
7. Line 323 – For 2 h... change to hours for consistency.
8. Line 400 – "Clair3with..." missing spacing.
9. Extended Data Fig. 5c and d – X-axis labels are "iPSCs chromosomes" and "iPSC chromosomes" respectively. Please change to keep consistency.

Reviewer #2 (Remarks to the Author):

Summary

=====

In this manuscript, Schmidt et al present a really interesting study on an approach they developed to enrich telomeric repeats for long-read sequencing. As highlighted in the introduction of the authors manuscript, there have been some related studies to enrich and sequence telomeres in yeast with nanopore sequencing, and in human with PacBio HiFi sequencing. However, it is my view that these prior studies do not reduce interest in the authors' work, but rather highlight the significant interest that the telomere field has in developing and applying methods to study telomeres with long-read sequencing. It is therefore my view that the work would be of significant interest to both the telomere and genomics community.

A key strength of the authors' study is the application of the method they have developed to a wide range of contexts where telomere length is expected to shorten (e.g. after multiple population doubling, in individuals of increasing age) and in other contexts where telomere length is known to lengthen (e.g. generation of iPSCs). These extensive validation efforts therefore convince me that the approach the authors developed can be used to track both increases and decreases in bulk telomere length. However, I do have some concerns about the authors claim and conclusion about the study of chromosome arm-specific telomere length in samples where a complete genome assembly is

unavailable (e.g. IMR90 , fibroblast, iPSCs, etc. in the authors's manuscript). The claims and conclusions that the authors made about arm-level telomere length therefore needs to be more extensively substantiated, or needs to be moderated.

Major point

=====

1) A major concern I have relates to the ability of the author's approach to accurately assign the telomeric reads to the correct chromosomal arms in samples without a complete reference genome. In the context of the HG002 sample where a complete reference genome of the sample is available (HG002 v0.7), telomeric reads can be assigned to the correct chromosomal arm reasonably well. Indeed, if we were to look at Extended Data Fig 1f where the authors had mapped the telomeric reads to the HG002 reference genome, a more or less even coverage of telomeric reads was observed (except for the acrocentric 13p, 21p, 22p), suggesting that there is minimal bias in assignment of telomeric reads. However, this is not the case where a complete reference genome is not available (e.g. IMR90, fibroblast, iPSCs, etc.). In these cases, we see a high degree of heterogeneity of coverage between chromosomal arms (e.g. ~2x higher coverage for 9p, 9q, 16p, and lower coverage for 5p, 1q, 3q for IMR90 in Extended Data Fig 2b), which suggests that telomeric reads for a significant fraction of chromosomal arms are likely misassigned. I therefore have significant concerns about the authors ability to assign the telomeric long-reads to the correct chromosomal arms for these other samples. Thus, the authors should provide evidence to suggest that telomeric reads can be accurately assigned to the correct chromosomal arms in these samples without a matched reference genome, or restrict their analysis to a subset of these arms for which telomeric reads can be accurately assigned and at the same time moderate their conclusion about the analysis of arm-level telomere length.

2) I think the readers would naturally be interested in the performance of Telo-seq, which the authors developed, versus other approaches to enrich and sequence telomeres with long-reads. In my view, it would be unreasonable to ask the authors to establish other methods and perform a head-to-head comparison. That said, I think it would be quite helpful if the authors can calculate and present some metrics from the sequencing data that they had already generated to help the readers better assess the performance of each of these methods. Some information which I think will be helpful to provide are:

a. Total number of reads in each sequencing run, number of telomeric reads, number of non-telomeric reads, etc.

b. For the telomeric reads, it would be helpful to provide a breakdown of read that falls into each of the categories depicted in Extended Data Fig 1b. The number of telomeric reads from G- vs. C-strand. Additionally, based on the Telo-seq oligos provided in Supplementary Table 7, it seems like the ends of the telomeres in the long-read would be marked by a "AGCAAT" sequence. What fraction of telomeric reads contains this tag sequence? How many telomeric reads contain both the tag sequence and subtelomeric sequences?

c. It may also be helpful to provide a brief description of the non-telomeric reads in these sequencing libraries (e.g. which part of the genome did they come from?).

3) The authors adopted a restriction digestion-based approach to enrich telomeric sequences in Telo-seq. Thus, in principle, each telomeric long read should start at a restriction site in the subtelomere and terminate at the end of the telomere. I therefore wonder if the same restriction site was used across all telomeric reads from the same chromosomal arm (i.e. the telomeric reads all end at the same restriction site)? Was this restriction site the closest possible restriction site to the telomeres? If not, does this indicate that digestion of DNA in the author's protocol is incomplete, or that these sites are blocked by DNA modifications? If there is heterogeneity in the restriction site used for the same chromosomal arm, is there a difference in telomere length measurements when different restriction cut sites are used?

4) Telo-seq seems to only target the "C-strand" of the telomeres (Fig 1a and line 81). As such, we should only see telomeric reads from the "C-strand" of the telomeres from nanopore sequencing with Telo-seq. What is the fraction of reads that were captured from the "C-strand" vs. the "G-strand" of

telomeres? I think this is a helpful metric to present as it can help the readers better assess the quality of the enrichment process.

5) A common issue associated with the assessment of telomere length with long-read sequencing is the establishment of the telomere-subtelomere boundary. Specifically, the presence of interstitial telomeric repeats and telomere variants in the subtelomeres can make the establishment of this boundary rather challenging for some chromosomal arms. To address this issue, the authors had concatenated these telomeric segments into a unified sequence if the breakpoints did not exceed 250 basepairs (line 374 – 377). While I don't think there is a consensus in the field as to how one should define the telomere-subtelomere boundary, it would be helpful if the authors can explain the rationale behind their approach. Additionally, I would presume that some telomeric long-reads might capture a longer subtelomere of the same chromosomal arm, and therefore cause more segments of telomeric repeats to be included in the calculation of telomere length. Would this then lead to differences in telomere length measurement for the same chromosomal arm even though the length of telomeric repeats on the most terminal segment is the same?

Minor Point

=====

1) It appears that several authors on the manuscript are employees of Oxford Nanopore. While I do not think their affiliation with Oxford Nanopore will affect the integrity of the study, this should be fully disclosed within the text. However, I do not see a "Conflict of interest" statement in the submitted manuscript.

2) There seems to be significant variation in the number of reads observed in each of the libraries (e.g. ~3-5k reads for iPSCs in Supplementary Table 4, ~15-50k reads for fibroblasts in Supplementary Table 3 even though the authors showed that these fibroblasts have shorter telomeres than iPSCs, and ~20-30k reads for IMR90 in Supplementary Table 2.). What is the cause of this variation, and does it have an impact on the telomere length measurements by Telo-seq?

3) Can the authors perform a densitometric analysis on their TRF result in Figure 1b and compare the distribution to that observed with Telo-seq in Figure 1d (akin to Figure 2B in PMID: 34702734)?

4) Lines 205-207 – This sentence can be written slightly clearer to emphasize the earlier study on heterogeneous telomere length in ALT cells. My initial impression was that this was an entirely new discovery from the authors' work.

5) The acrocentric arms (13p,14p,15p,21p,22p) are known to undergo frequent recombination (<https://www.nature.com/articles/s41586-023-05976-y>). It may therefore be difficult to distinguish telomeres that originate from each of these arms. This should be made clear in the figures and text.

6) Line 363-364: Information on the bonito model used for calling telomere repeats is missing (e.g. version of the model, where to get it, how it was trained/obtained etc.)

7) I note that the GridION nanopore sequencer was used in the author's study. Given the wider accessibility of the P2/P2-solo sequencer, readers may be interested in the performance of Telo-seq on a PromethION flow cell. It would therefore be helpful to include this information in the manuscript if the authors have it on hand, though this is not absolutely necessary.

8) Extended Figure 1d – WGS and Telo-seq was performed on the same sample. Do both approaches give rise to similar telomere length measurements?

9) Figure 2e – This is quite interesting. A nice and almost linear correlation of mean telomere length and population doubling was observed using Telo-seq.

10) Line 371 – A minimum length of 500 bp of unique subtelomeric sequences was required for the reads to be considered. Is 500 bp sufficient to uniquely distinguish one chromosomal arm from another?

11) Line 517, 528, 543, 522 – Fig S4-S7 should be labelled as Extended Data Fig.

12) Supplementary Tables – Does "No. of reads" represents the total number of reads in the library, the number of telomeric reads, or the number of filtered telomeric reads (based on criteria in line 366-371)? Can information on the total number of reads in each library be provided? Are the mean, std dev, median, IQR, values etc. calculated based on the length of telomeric repeats, or the length of the whole read?

Reviewer #3 (Remarks to the Author):

In this manuscript, "High resolution long-read telomere sequencing reveals dynamic mechanisms in aging and cancer," Schmidt, Tyer, and Rugh et al., present their new method, "Telo-seq," which is built to resolve bulk, chromosome arm-specific and allele-specific human telomere lengths using Oxford Nanopore Technologies' native long-read sequencing. Telo-seq is a new method that uses telorette-based telomere adapters that are annealed to the G-overhang and ligated to the C-strand, digested with the blunt end restriction enzyme EcoRV, and includes a step to reduce concatemer ligation (dA-tailing). This method was shown to resolve telomere shortening in five population-doubling increments and revealed intrasample, chromosome arm-specific, allele-specific telomere length heterogeneity. Telo-seq showed some ability to reliably discriminate between telomerase- and ALT-positive cancer cell lines.

They compared their Telo-seq bulk TL measurements of two independent human cell line replicates to TRF results, and they were in line. Both also showed very similar subtelomere and telomere length distributions. They conclude that Telo-seq can reproducibly measure bulk and chromosome arm-specific TL dynamics in human cells, and resolve telomere shortening rates.

In human donor-derived fibroblasts, telomeres shortened with increasing age. Further, some chromosome arms consistently had shorter or longer telomeres, relative to the mean telomere length, raising the possibility that conserved chromosome arm-specific features influence telomere length, which are currently unknown. Telo-seq also allows analysis of higher resolution allele-specific telomere length.

Thus, Telo-seq is a novel tool to study telomere biology during development, aging, and cancer at unprecedented resolution, and will be useful for the field. While the paper is generally sound, a few suggestions remain that would benefit from clarification:

1. They mention the A-tailed step to reduce concatemers, but don't show much data on this, and it would be good to make sure there are not artifacts induced from the protocol.
2. Related to #1, they mention using a customized Bonito telomere model, but previous studies have shown that each version of the ONT software can introduce entirely new repeat calls and false positives, and it would be good to see a comparison (and ideally replication) of their results on a more recent version of the Bonito software.
3. Their adapters used the canonical telomeric repeats and got a 75-fold increase in enrichment, but would a mixture of some of the known, non-canonical sequences included in the adapters help yield? Did the authors try this?
4. Related to #3, did the authors observe any non-canonical telomere variants? These have been documented in the literature and would be expected here as well.
5. They showed a general trend of shorter telomeres with age, but there was a 82-year-old donor that was an outlier; was there anything clinically or health-wise unique about this individual that could explain this difference?
6. The authors describe the methylation patterns in the CpG islands from their data; but did they look at the prediction of other base modifications? This might be interesting to explore, in this, or a follow-up paper.
7. The software referred to in this paper should be made available.
8. The raw data used for the paper is not available, making it impossible to validate the findings, and this too needs to be made available.
9. They conclude that, upon induced pluripotency, shorter telomeres are preferentially elongated by telomerase; however, the overall order of chromosome arm-specific telomere length remained conserved between fibroblasts and iPSCs. This should be expanded upon in the discussion or more in

the results.

10. They propose that Telo-seq is an effective method to evaluate the TMM of cancer cells, as it recapitulates the differences in TL distributions in telomerase vs ALT+ cells. How well would this expand to other cell lines and genomes?

11. Importance/significance: This study highlights the potential of human telomere long-read sequencing and sets the stage to investigate human telomere dynamics in unprecedented detail during development, aging and disease. This could be discussed more.

12. They have referenced some prior work in telomere sequencing, including data from the Telomere-to-Telomere (T2T), and Genome in the Bottle Consortia, but there are other papers which have already used ONT and long-read data to map telomere length and variation, which should be cited, including: <https://pubmed.ncbi.nlm.nih.gov/33242406/> <https://pubmed.ncbi.nlm.nih.gov/33242411/>

Response to referees NCOMMS-23-51397-T

Reviewer #1 (Remarks to the Author):

Tobias T et al have demonstrated the capability to measure telomere length using Oxford Nanopore long-read sequencing platform. The authors also design a method (Telo-seq) to enrich for telomere containing genomic DNA, which resulted in more than 75-fold increase in telomeric read output comparing to whole genome sequencing. A customized Bonito telomere model was developed for basecalling and sequencing data analysis, which should be useful for scientists studying telomere length in cancer and aging. Allele-specific telomere length can be measured for all 44 chromosomal ends excluding the sex-determinant chromosomes, showing potential chromosome arm-specific length heterogeneity. In addition to these findings, the authors also apply this new telomere sequencing method to successfully differentiate TERT+ cancer cells and ALT+ cancer cells by measuring the dispersion of their telomere length distribution.

Overall, the results are clear, and the method is interesting. However, the current manuscript is missing key details for comprehensive evaluation of the data, which may limit this new approach to recover accurate mean telomere lengths as well as allele-specific telomere lengths.

Response: We thank the Referee for their support and constructive comments, which are making this a better manuscript. Please find detailed answers to the individual points of criticism below.

Major comments:

1. The Telo-seq method adapted a similar strategy as STELA by ligating telorette-based telomere adapters to the C strand of telomeric ends. This will inadvertently result in loss of information for telomere G tail overhangs, which is known to vary from 100-300 nucleotides in length in human cells. Therefore, the newly established Telo-seq may result in underestimation of actual telomere length. Given the peripheral blood mononuclear cells telomere shortening rate is about 40-50 nts per year, whether this method can be utilized for accurate measurement of telomere length shortening rate in general population remains to be validated.

Response: We agree with the reviewer that Telo-seq uses like STELA ¹ and TESLA ² telorette-based telomere adapters and that Telo-seq is therefore not providing any information about the telomeric G-rich overhang length. The same argument is true for STELA and TESLA, which are both well-established methods in the telomere field to measure telomere length. Thus, if the telomeric G-rich overhang length is critical for a specific research question, alternative methods to STELA, TESLA and Telo-seq have to be applied.

The telomere shortening rate of ~30-50 nt in adult peripheral blood mononuclear cells have been determined based on a larger cohort of individuals. First, the individual's telomere length was determined with TRF ³⁻⁵ or flow FISH ⁶, for example. Second, the determined telomere length was plotted against the donor age and linear regression analysis performed. Our data on IMR90 ^{E6E7} fibroblasts at different population doublings suggest that Telo-seq can differentiate telomere length in samples 5 population doublings apart (Fig. 2). Given that Telo-seq has similar or higher resolution than TRF and age-dependent Telo-seq-based telomere length decrease is also observed in the fibroblasts of the investigated aging cohort (Fig. 4c), we therefore postulate that, comparable to previous work, an averaged peripheral blood mononuclear cell telomere shortening rate per year from a cohort can be determined with Telo-seq.

2. The customized Bonito telomere model that was developed for basecalling in Oxford Nanopore sequencing data analysis will be valuable to the research interest of telomere length in cancer

and aging. Please explain the detailed training process for this new model. Please also describe the detailed normalization steps that have been applied to read coverage.

Response: We thank the reviewer for highlighting this point. We have expanded the method section in the revised manuscript.

3. Please show the raw read coverage in all figures, which is essential to judge the accuracy of telomere length measurement.

Response: We appreciate the reviewer's comment. In the revised manuscript, we have added the number of telomeric reads corresponding to each bulk telomere length analysis figure panel. A summary for each flow cell and sample including number of telomeric reads is shown in the Supplementary Table 1, 3, 7, 8, 10, 11. Due to figure font size limitations we have added additional Supplementary Tables 2, 4 5, 7, and 9 to the revised manuscript including detailed information about chromosome arm-specific telomere length analysis, including number of telomeric reads per chromosome arm. The revised figure legends link to the corresponding Supplementary tables.

4. The telomere length estimated from HG002 Chr. 21p before haplotype phasing is above 9kb (Fig. 1e and lines 107-108). However, the lengths were estimated to be 4-5kb and 10-11kb for maternal and paternal alleles separately (Extended Data Fig. 4a). So, if the two alleles have been sequenced in similar depth, the telomere length of Chr. 21p should be much smaller than 9kb. Chr.1p has the same problem. Could you elaborate more regarding these differences. In addition, the allele-specific telomere lengths for paternal Chr. 13p and Chr. 22p are missing in Extended Data Fig. 4a as well.

Response: The reviewer's assumption that the differences between the chromosome arm-specific telomere length of HG002 and the two corresponding alleles can be due to different allelic coverages (Supplementary Table 2 and 5) is correct. This is likely the reason why the 1p chromosome arm-specific telomere length is closer to the maternal allele-specific telomere length (610 maternal and 337 paternal reads) (Supplementary Table 2 and 5). For Chr. 21p, both alleles have roughly the same number of reads (133 maternal and 113 paternal). However, we like to note that the boxplots show the median telomere length for each chromosome arm. Therefore, in the cases, in which the maternal and paternal telomere length are substantially different from each other and form distinct telomere length populations, even with same allelic coverage, the median will be closer to one allelic population than to the other one and likely be different than the mean telomere length of both populations. As suggested by the reviewer in major comment 3, we have added the underlying number of reads to Supplementary Table in the revised manuscript.

The paternal HG002 alleles of chromosome arm 13p and 22p are not shown in Fig. 3, because the mapping quality of both paternal alleles are below the applied mapping quality filter of 5. We have added a comment in the figure legend to explain why only the telomere length of the maternal 13p and 22p alleles are shown. We have also added a plot of the mapping quality per allele in HG002 in Extended Data Fig. 4a of the revised manuscript.

5. In Extended Fig. 4a, it is very clear that both paternal and maternal allele in HG002 chr18q have two sets of reads in it: One is about 4-6kb, the other is about 8-10kb. Why the two alleles in this specific arm exhibit 2 sets of telomeres with different length?

Response: We agree with the reviewer that for both HG002 chromosome 18q alleles the majority of reads cluster around 8-10 kb and there is an accumulation of reads at 4-6 kb. The appearance of shorter telomeres could be indicative of a subpopulation of cells that have acquired shorter telomeres at chromosome 18q, for example due to rapid telomere shortening. The relatively long

telomere length of 4-6 kb will likely not interfere with growth as other HG002 chromosome arms have shorter telomere length.

6. It is known that telomere length is reversely correlated with age. However, the authors have identified an 82-year-old individual who has abnormal longer telomere compared to the other individuals in his/her age group (Fig. 3a). How many replicates have been sequenced using Oxford Nanopore platform for each individual? Have the authors compared using the traditional TRF method to measure the telomere length in these eight individuals to validate the accuracy of their telomere length estimation using Oxford Nanopore platform.

Response: The reviewer stresses an important point with this comment. Previous work has shown that the telomere length of individuals in the population even in the same age group is very heterogeneous⁶⁻⁸. Whereas all of these studies observe a reverse correlation of telomere length with age, there are also always individuals reported that have abnormal long telomeres relative to their respective age group, but also in comparison to individuals several decades younger. Therefore, we think that the 82-year-old individual present in our small aging cohort is indeed one of these individuals, who has relatively long telomeres for their age.

As we have shown reproducibility of Telo-seq results for HG002 and cancer samples (Supplementary Table 1, 11), we have sequenced one replicate per sample for the aging cohort samples (Supplementary Table 6). We have compared TRF and Telo-seq telomere length for the HG002, IMR90^{E6E7} and cancer samples and got overall consistent results between both methods (Fig. 1, 2, 5). Due to limited access to sample, we could not run a TRF on the eight individuals in the aging cohort.

7. While Telo-seq may be able to distinguish TERT+ cancer cells and ALT+ cancer cells, the telomere length distribution in ALT cells obtained from Nanopore sequencing seem to be much shorter than the traditional teloblot (compare Fig. 4b and Fig. 4a). What is the potential cause for such discrepancy? The intense high molecular weight fractions in ALT samples seem to indicate incomplete genomic DNA digestion as well.

Response: We agree with the reviewer that the telomere length distribution of the ALT+ cancer cells seem to be longer in the TRF analysis than in Telo-seq.

Based on the methodical differences there are several explanations:

- 1) For the TRF analysis, high-molecular weight genomic DNA is digested with restriction enzymes (AluI/MboI in our protocol) that frequently cut outside the telomeric motif. So telomeric TRF fragments contain telomeric repeats and adjacent subtelomeric sequence. Therefore, the detected telomere length in TRF is the sum of the telomere and the remaining subtelomere present in the fragment, whereas for Telo-seq we are only plotting the telomere length. Longer estimated telomere length based on TRF relative to telomere length determined by long-read sequencing have been also reported in a recent study for mouse cells⁹.
- 2) TRF uses a telomeric probe to stain telomeric repeat containing fragments. As consequence, fragments with more telomeric repeats (longer telomeres) will allow more probe binding, resulting in higher intensity for longer than for shorter telomeres. For example, HT-29 and SK-N-AS have more intense telomere distribution than Calu-3 (Fig. 5a). Thus, shorter telomeres appear weaker on a TRF than longer telomeres. Contrary, in Telo-seq plots every telomeric read will be equally visualized, independently of its length.
- 3) Fragments in a TRF analysis are separated on an agarose gel which runs non-linear. Smaller fragments will be separated on a larger distance. Consequently, longer fragments run closer to each other and are relatively compressed compared to shorter fragments.

Thus, in addition to the probe binding, the gel running behavior is contributing to the visual impression of higher fragments being more abundant than shorter fragments, especially in samples with very heterogenous telomere length distributions like ALT⁺ cancer cells.

- 4) In TRF, the probe is staining any fragment which contains telomeric repeats, including interstitial telomeric repeats. These interstitial telomeric repeats can be generated in a chromosome fusion event and will lead to distinct bands in a TRF, for example. In contrast, Telo-seq captures the telomeric ends by using telorette-based adapters specific for the telomeric overhang and is therefore very unlikely to sequence these interstitial telomeric repeats. In highly rearranged and genetically unstable genomes like cancer genomes, especially ALT⁺ cancer cells, these interstitial telomeric repeats may be more common.

Taken together, we like to argue that these methodological differences contribute to and explain the impression that ALT cells seem to have longer telomeres based on TRF analysis.

Incomplete digestion of genomic DNA could be an explanation for high molecular weight fraction. However, based on the SybrGold staining that we have done for the gel prior to Southern blotting (Reviewer Fig. 1), we do not think that the high molecular weight fragments are due to incomplete digestion.

Reviewing Fig. 1. TRF analysis of cancer cell lines. AluI/MboI digested genomic DNA of 5 TERT⁺ and ALT⁺ cancer cell lines were separated on a 0.7% agarose gel. Left) agarose gel stained with Sybr Gold (1:10,000 in 0.5x TBE, 40 min, RT). Right) TRF using TelG telomeric probe (as shown in Fig. 5a)

8. The author indicated that “In ALT⁺ cells, we could measure telomeres from 47 bp to 134.7 kb in length.” The 47 bp telomere seem to be too short. How often does the sample show telomere reads with such short length (percentage)?

It is highly possible that such fragments (47 bp) may be resulted from random shearing of genomic DNA during genomic DNA extraction.

Response: The very short telomere lengths, below 50 and 100 bp, are generally very infrequent (Reviewer Fig. 2). Only the ALT⁺ cell line SK-LU-1 with the shortest telomere length distribution had a significant accumulation of very short telomeres below 100 bp (~4%) (Reviewer Fig. 2 c). Telomeres between 1-50 bp are very infrequent (Reviewer Fig. 2d).

We cannot completely rule out that some of the reads with very short telomeres are due to random sharing during DNA prep or handling. However, as we are using a probe against the overhang to introduce the sequencing adapter, we consider it as rather unlikely that the randomly shared DNA will produce an overhang that is compatible with the probe. Further, as postulated for cells without an active telomere maintenance mechanism, short telomeres (<1 kb) are accumulating in IMR90^{E6E7} with increasing population doubling (Fig. 2d, Reviewer Fig. 2), suggesting that the short telomeres are present in these samples and not (exclusively) a product of sample preparation (Reviewer Fig. 2). Finally, independently to us, the Artandi lab also reports the presence of short telomeres in their preprint on human telomere nanopore sequencing (<https://doi.org/10.1101/2023.11.29.569263>).

Reviewer Fig. 2. Distribution of telomere length in cancer and IMR90^{E6E7}. **a**, binned telomere length in base pairs (bp) of cancer cell lines. **b**, binned telomere length IMR90^{E6E7} at indicated population doublings (PD). **c**, bar graph of the percentage of telomeres below 100 bp. Number of telomeric reads with a telomere length between 1-100 bp is shown above the bar. **d**, bar graph percentage of telomeres below 50 bp. Number of telomeric reads with a telomere length between 1-50 bp is shown above the bar.

9. The Allele-specific telomere length measurement is certainly one of the most interesting results in this manuscript.

Response: We thank the reviewer for this comment that encouraged us to move the mapped phased HG002 telomere length (submitted manuscript Extended Data Fig. 4a) as independent main text Figure 3 in the revised manuscript. We moved the paragraph about allele specific telomere length assignment up in the revised manuscript text right after the “Telo-seq resolves telomere shortening” paragraph.

While the authors showcase the capability to measure the telomere length of all 88 alleles, whether they are accurate remains a major question. For example: While the author highlighted their ability to accurately measure telomere shortening rate (average 39 bp per PD) in IMR90E6E7 cells in serial passage in vitro, the allele-specific telomere length shortening seems to have a lot of heterogeneity. In many instances (Fig. 2f, Chr. 22p PD106.1; Chr. 14p PD106.1; Chr 12q PD106.1, etc) the allele-specific telomere lengths in the late passage are much longer than IMR90E6E7 in earlier passages. These discrepancies of allele-specific telomere length bring a lot of doubt on the accuracy of allele-specific telomere length measurement using current method.

Response: As the reviewer has correctly pointed out, the median chromosome arm-specific telomere length is increasing in some chromosome arms especially in the later passages (PD106). However, we like to note that these cells (IMR90^{E6E7} PD106) are encountering replicative crisis¹⁰⁻¹². Crisis, or mortality stage 2, is a telomere-dependent proliferation barrier, in which most cells die an autophagy-dependent, innate immunity-driven cell death^{13,14}. Thus, cells with dysfunctional telomeres will be efficiently removed from the population. As telomere length in a cellular population is distributed around an averaged telomere length¹⁵, there will be some cells with overall shorter, or with short telomeres, at only some chromosome arms. We postulate that this subpopulation of cells will experience crisis earlier than the rest of the population. Thus, our interpretation of the results is that there is counterselection against the subpopulations of cells that have very short bulk telomeres, or very short telomeres, at individual chromosome arms in cells approaching crisis as they are experiencing crisis earlier and are removed from the population. Consequently, under these selection conditions, clones with longer telomeres will outgrow the ones with shorter ones. For the last sampling point (PD106), but likely also for cells that bypass the senescence plateau (PD96), there are two processes affecting the chromosome arm-specific telomere length measurements: telomere shortening and selection for a population of cells with telomeres long enough to protect chromosome ends. Consequently, we have excluded the two last timepoints for the calculation of chromosome arm-specific shortening rates (Extended Data Fig. 3c).

In the revised manuscript we have included additional benchmarking figures based on HG002 Telo-seq analysis (Extended Data Fig. 1 and 2), updated the chromosome arm-specific telomere length figures using a mapping quality of 20 and discussed current limitations of chromosome arm-specific telomere length assignment in the “Discussion”.

In addition, only the chromosomal end specific telomere lengths are presented, but not the allele-specific telomere lengths of IMR90E6E7 cells are shown for each PD? Given the differences of allele-specific telomere length (Extended Data Fig. 4c), plotting the allele-specific telomere shortening rate in Fig. 2f will be important.

Response: We agree with the reviewer that allele-specific shortening rates would be very interesting. However, resolving allele-specific telomere length is challenging and, given the

heterogeneity, requires a high coverage per chromosome arm. Phasing quality also depends on the reference genome. We present a de novo approach (Extended Data Fig. 4b,c); however, as discussed in the manuscript, this approach is not optimal, and we cannot resolve all alleles for every chromosome arm. One key issue is that for IMR90^{E6E7} as well as for all other samples in this study, except for HG002, there is no phased reference genome and parental genome information available. Given these challenges we do not feel confident in calculating allele-specific shortening rate at this point.

10. By profiling allele-specific telomere length using donor-derived fibroblasts, the authors have indicated that some chromosome arms consistently had shorter or longer telomeres. They further proposed that there are potential conserved chromosome arm-specific features influence telomere length. Similar results have been reported previously (Londono-Vallejo J.A. NAR 2001; Martens U.M. Nature Gen. 1998; Graakjaer J. Mech. Ageing Dev. 2003 etc) using other telomere length measurement methods, such as FISH. However, there are no consistency on which chromosomal end has the shortest or the longest telomere.

Response: We thank the reviewer for this comment. Given the high heterogeneity of the telomere length on the chromosome arm level and the relatively small cohort size (≤ 20 individuals in previous^{16,17} and 9 in our work), we believe that to comprehensively address the question whether specific telomeres at some chromosome arms are consistently shorter or longer requires a large, diverse, and well-controlled cohort. Nevertheless, despite some discrepancy with and within previous data, the five shortest (chromosome 21p, 19p, 19q, 16q, 9q) and longest telomeres (chromosome 18q, 12q, 3p, 4q, 5p) identified in our cohort are also detected to be shorter and longer in two previous studies^{16,17}. We have expanded this point in the discussion of the revised manuscript.

Similar results were observed in Extended Data Fig. 4a: most of the two alleles from the same chromosomal end seems to have vast different telomere length. The allele-specific telomere length for the eight samples should be shown in Extended data Fig. 3c. In summary, whether there is conserved chromosome arm-specific features influence telomere length remains to be addressed.

Response: Telomere phasing without the sample's phased reference genome is difficult and our de novo phasing approach frequently results in alleles and chromosome arms that cannot be resolved (Extended Data Fig. 4b,c). In addition, we do unfortunately not have enough coverage to present allele-specific telomere length for the eight fibroblasts.

We agree with the reviewer that it is an open question, whether conserved chromosome arm-specific features exist that influence human telomere length. Our analysis is limited to only 9 fibroblast samples. Whether there are chromosome arm-specific features that influence telomere length, and which chromosome arms have the longest/shortest telomeres has to be determined in future studies. To answer these questions comprehensively, analysis of a much larger, well-controlled, diverse cohort is required. This is beyond the scope of this manuscript.

11. What is the meaning of the methylation data? There is a lot of heterogeneity, which may be due to the sequencing method itself or read depth. Is the methylation results shown in each panel in extended data Fig. 6 average of multiple reads from the same allele? The results have shown that ALT+ cancer cell lines were observed to be hypomethylated compared to TERT+ cancer cell lines. However, it is also mentioned that not all the arms were equally hypomethylated. Can the authors specifically list out the hypomethylated/hypermethylated chromosome arms? And within the differentially methylated arms, what are the differentially methylated regions (DMR)? Does the

ALT+ cell lines have similar DMR compared to TERT+ cell lines? Furthermore, it is also important to compare the methylation pattern of cancer cells with normal cells (HG002 and IMR90).

Response: The advantage of native nanopore sequencing and Telo-seq is that, in addition to the sequence information and telomere length, the methylation status of the CpG sites closest to the telomeres can be obtained. We agree there is inter- and intra-sample heterogeneity. However, to our knowledge this type of analysis has not been done in this detail before and highlights which type of questions can be answered by Telo-seq.

We thank the reviewer for suggesting a summary about the methylation status of the CpG sites closest to the telomeres in controls and cancer cell lines. This will provide a fast overview and facilitate the search for the chromosome arms where the methylation status differs from controls. We have generated a heatmap that shows the averaged methylation of every chromosome arm for the indicated sample and added it to Extended Data Fig. 7.

Minor comments:

1. As indicated by the authors, Telo-seq resulted in “increased telomeric reads 75-fold relative to whole-genome sequencing (Extended Data Fig. 1d).” Such enrichment may pose a problem for high-throughput telomere length application as the author has pointed out that “the telomeric content of human diploid cells is approximately only 0.015% of the total genome.” The number of samples that can be applied in single Nanopore flow cell should be mentioned.

Response: In this study we used one sample per MinION flow cell to establish the method. We added a sentence in the “Telo-seq” methods section. We currently do not multiplex samples. However, the Artandi lab describe multiplexing using the bigger PromethION flow cells in their preprint (<https://doi.org/10.1101/2023.11.29.569263>).

2. Are the two replicates of HG002 shown in Fig. 1 obtained in the same or different Nanopore flow cell?

Response: The two replicates were run on two different flow cells. The underlying data of Fig. 1 are listed in Supplementary Table 1 and 2.

3. In Extended Data Fig. 2c, the shortening rate of 20q, 21p, 1q, and 1p are not significant. Is that because of low sequencing depth of these arms? If so, this analysis can be refined by including more reads.

Response: The chromosome arms with non-significant p-values have one or more time points in which the median telomere length is similar or larger than in the previous time point. Low read coverage, which makes the median and mean less stable (Extended Data Fig. 1h), is likely the major reason for the non-significant p-values of chromosome 20q, 21p and 5p. We do not have a good explanation for chromosome 1p. Coverage is not an issue for chromosome 1p and telomere length decreases in later passages (PD96 and PD106). For the chromosome arm-specific shortening calculation, we have excluded the two latest time points to avoid clonal selection (please see also response to reviewer comment 9). Chromosome 1q has a p-value < 0.05.

4. Line 174, the author mentioned that pangenome reference could result in higher alignment quality compared to haploid reference genome. Please give a quantitative measurement of higher quality.

Response: We have updated the main text and changed the referred sentence to “Secondly, using a custom pangenome reference leads to higher quality alignments compared to solely relying on a single reference, especially when dealing with a diverse patient population where an exact reference is not available.”

5. Extended Data Figure legends: Please use either “Extended Data Fig. 1-7” or “Fig. S1-7” to be consistent.

We have changed the labeling of the Extended Data Figures to “Extended Data Fig. 1-7”.

6. Line 259 – Fig 3d missing a “.”, Fig. 3d.

We have added the “.”.

7. Line 323 – For 2 h... change to hours for consistency.

We have changed the “2 h” to “2 hours”.

8. Line 400 – “Clair3with...” missing spacing.

We have added the missing spacing.

9. Extended Data Fig. 5c and d – X-axis labels are “iPSCs chromosomes” and “iPSC chromosomes” respectively. Please change to keep consistency.

We have changed the X-axis label in Extended Data Fig. 5c to “iPSC chromosomes”.

Reviewer #2 (Remarks to the Author):

Summary

=====

In this manuscript, Schmidt et al present a really interesting study on an approach they developed to enriched telomeric repeats for long-read sequencing. As highlighted in the introduction of the authors manuscript, there have been some related studies to enrich and sequence telomeres in yeast with nanopore sequencing, and in human with PacBio HiFi sequencing. However, it is my view that these prior studies do not reduce interest in the authors' work, but rather highlight the significant interest that the telomere field has in developing and applying methods to study telomeres with long-read sequencing. It is therefore my view that the work would be of significant interest to both the telomere and genomics community.

A key strength of the authors' study is the application of the method they have developed to a wide range of contexts where telomere length is expected to shorten (e.g. after multiple population doubling, in individuals of increasing age) and in other contexts where telomere length is known to lengthen (e.g. generation of iPSCs). These extensive validation efforts therefore convince me that the approach the authors developed can be used to track both increases and decreases in bulk telomere length. However, I do have some concerns about the authors claim and conclusion about the study of chromosome arm-specific telomere length in samples where a complete genome assembly is unavailable (e.g. IMR90, fibroblast, iPSCs, etc. in the authors's manuscript). The claims and conclusions that the authors made about arm-level telomere length therefore needs to be more extensively substantiated, or needs to be moderated.

Response: We thank the Referee for their generous support and constructive comments, which are making this a better manuscript. Please find detailed answers to the individual points of criticism below.

Major point

=====

1) A major concern I have relates to the ability of the author's approach to accurately assign the telomeric reads to the correct chromosomal arms in samples without a complete reference genome. In the context of the HG002 sample where a complete reference genome of the sample is available (HG002 v0.7), telomeric reads can be assigned to the correct chromosomal arm reasonably well. Indeed, if we were to look at Extended Data Fig 1f where the authors had mapped the telomeric reads to the HG002 reference genome, a more or less even coverage of telomeric reads was observed (except for the acrocentric 13p, 21p, 22p), suggesting that there is minimal bias in assignment of telomeric reads. However, this is not the case where a complete reference genome is not available (e.g. IMR90, fibroblast, iPSCs, etc.). In these cases, we see a high degree of heterogeneity of coverage between chromosomal arms (e.g. ~2x higher coverage for 9p, 9q, 16p, and lower coverage for 5p, 1q, 3q for IMR90 in Extended Data Fig 2b), which suggests that telomeric reads for a significant fraction of chromosomal arms are likely misassigned. I therefore have significant concerns about the authors ability to assign the telomeric long-reads to the correct chromosomal arms for these other samples. Thus, the authors should provide evidence to suggest that telomeric reads can be accurately assigned to the correct chromosomal arms in these samples without a matched reference genome, or restrict their analysis to a subset of these arms for which telomeric reads can be accurately assigned and at the same time moderate their conclusion about the analysis of arm-level telomere length.

Response: We thank the reviewer for this important comment. We have included additional benchmarking figures based on HG002 Telo-seq analysis to the revised manuscript (Extended Data Fig. 1 and 2) and updated the mapping quality for chromosome arm-specific telomere length assignment. Further, in the discussion of the revised manuscript, we now address limitations of the Telo-seq method in respect to chromosome arm-specific telomere length measurements for samples without a phased T2T reference genome. We have added the sentence “For limitations on Telo-seq chromosome arm-specific telomere length assignment see “Discussion”.” to the figure legends of the figures for which we show chromosome arm-specific telomere length analysis without matching T2T reference genome (Fig. 2f, Extended Data Fig. 5 and 6).

2) I think the readers would naturally be interested in the performance of Telo-seq, which the authors developed, versus other approaches to enrich and sequence telomeres with long-reads. In my view, it would be unreasonable to ask the authors to establish other methods and perform a head-to-head comparison. That said, I think it would be quite helpful if the authors can calculate and present some metrics from the sequencing data that they had already generated to help the readers better assess the performance of each of these methods. Some information which I think will be helpful to provide are:

a. Total number of reads in each sequencing run, number of telomeric reads, number of non-telomeric reads, etc.

Response: We thank for reviewer for the suggestion and have added this information to the Supplementary Tables.

b. For the telomeric reads, it would be helpful to provide a breakdown of read that falls into each of the categories depicted in Extended Data Fig 1b. The number of telomeric reads from G- vs. C-strand. Additionally, based on the Telo-seq oligos provided in Supplementary Table 7, it seems like the ends of the telomeres in the long-read would be marked by a “AGCAAT” sequence. What fraction of telomeric reads contains this tag sequence? How many telomeric reads contain both the tag sequence and subtelomeric sequences?

Response: We have added the information on how many telomeric strands are C-strand reads to the Supplementary Tables.

Out of 61,235 reads from HG002 that contained telomere motifs prior to read classification, we identified 130 reads that began with the “AGCAAT” sequence. 124 of these reads also include subtelomere sequence. Nonetheless, it is crucial to acknowledge that Bonito’s peak detection and trimming algorithm in signal space might have led to trimming of the beginnings of each read, potentially affecting our capacity to identify the “AGCAAT” sequence in the base space.

c. It may also be helpful to provide a brief description of the non-telomeric reads in these sequencing libraries (e.g. which part of the genome did they come from?).

Response: We thank the reviewer for this comment. We have aligned all HG002 reads to the HG002 reference genome. We show the binned coverage (bin size 2.5 kb) in the revised manuscript in the Extended Data Fig. 1g. Telo-seq reads are enriched at telomeres. Non-telomeric reads are randomly distributed throughout the genome.

3) The authors adopted a restriction digestion-based approach to enrich telomeric sequences in Telo-seq. Thus, in principle, each telomeric long read should start at a restriction site in the subtelomere and terminate at the end of the telomere. I therefore wonder if the same restriction

site was used across all telomeric reads from the same chromosomal arm (i.e. the telomeric reads all end at the same restriction site)? Was this restriction site the closest possible restriction site to the telomeres? If not, does this indicate that digestion of DNA in the author's protocol is incomplete, or that these sites are blocked by DNA modifications? If there is heterogeneity in the restriction site used for the same chromosomal arm, is there a difference in telomere length measurements when different restriction cut sites are used?

Response: In the Telo-seq protocol, we are sequencing from the end of the telomere into the subtelomere. Therefore, to determine the telomere length in a read the length of the subtelomere is neglectable, as long as enough non-telomeric sequence is present to define the subtelomere-telomere boundary. We have found that restriction enzyme digestion within the Telo-seq protocol increases the total number of telomeric reads that we obtained. For HG002, for which a phased reference genome exists, we generated an in-silico digest of the subtelomeres and compared the distribution to the HG002 subtelomere length obtained with Telo-seq (Extended Data Fig. 2 b). We do not observe a major difference between the Telo-seq read and in silico digested subtelomere length, suggesting that most fragments are cut at the predicted EcoRV site closest to the telomere.

For the reads that differ from the expected subtelomere length there are multiple reasons, as already suggested by the reviewer, including: 1) According to NEB, EcoRV can be blocked by some combinations of CpG islands. 2) There could be random breaks during the library preparation between the telomere-subtelomere boundary and the restriction cutting site. However, these alterations in subtelomeric sequence length will not affect the assignment.

4) Telo-seq seems to only target the "C-strand" of the telomeres (Fig 1a and line 81). As such, we should only see telomeric reads from the "C-strand" of the telomeres from nanopore sequencing with Telo-seq. What is the fraction of reads that were captured from the "C-strand" vs. the "G-strand" of telomeres? I think this is a helpful metric to present as it can help the readers better assess the quality of the enrichment process.

Response: We have added the information on how many of the telomeric reads are C-strand reads to the Supplementary Tables. In all runs, at least 98.39% of the telomeric reads correspond to the C-strand, with the exception of SK-LU-I for which 95.87% of telomeric reads were C-strand reads.

5) A common issue associated with the assessment of telomere length with long-read sequencing is the establishment of the telomere-subtelomere boundary. Specifically, the presence of interstitial telomeric repeats and telomere variants in the subtelomeres can make the establishment of this boundary rather challenging for some chromosomal arms. To address this issue, the authors had concatenated these telomeric segments into a unified sequence if the breakpoints did not exceed 250 basepairs (line 374 – 377). While I don't think there is a consensus in the field as to how one should define the telomere-subtelomere boundary, it would be helpful if the authors can explain the rationale behind their approach. Additionally, I would presume that some telomeric long-reads might capture a longer subtelomere of the same chromosomal arm, and therefore cause more segments of telomeric repeats to be included in the calculation of telomere length. Would this then lead to differences in telomere length measurement for the same chromosomal arm even though the length of telomeric repeats on the most terminal segment is the same?

Response: We agree with the reviewer that there is no consensus in the field for how to determine the subtelomere-telomere boundary and consequently telomere length. We decided to use the approach outlined in the methods, because it allows some flexibility for smaller insertions in the

telomeric track. We have expanded the rationale behind our approach in the method section of the revised manuscript.

In our experience, an increase in subtelomere length used does not lead to an increase in telomere length. To address whether varying subtelomere length influences telomere length determination, we made use of the paternal HG002 T2T reference genome. We extracted differently sized terminal fragments (7.5, 10, 15, 20, 25 and 50 kb) from chromosome ends (chromosome 1-5 p and q arm) and calculated telomere length of these fragments based on our telomere length determination criteria (Reviewer Fig. 3); these fragments share the same telomeric track (the telomeric sequence deposited in the reference genome) and only differ in the length of their subtelomeric track. We find identical telomere length, independent of the length of the subtelomeric track for all fragments of chromosome 2p, 3q, 4p, 5p and 5q. For the other chromosome arm (1p, 1q, 2q, 4q) the telomere length increases in length with increasing fragment size before plateauing. This can be explained by the fact that these chromosomes have longer telomeres than the total length of the smaller fragments. These shorter fragments consist entirely of telomeres and lack any subtelomeric sequences (colored red in Reviewer Fig. 3). Thus, our analysis suggests that telomere length assignment is reproducible and independent of subtelomere length.

Reviewer Fig. 3. Telomere length determination is independent of subtelomere length. 7.5, 10, 15, 20, 25 and 50 kb terminal fragments of the chromosome 1-5 of paternal HG002 T2T reference genome were extracted, and telomere length determined. Telomere length in kilobase pairs (kp) is shown for each fragment length for the indicated chromosome arms. Fragments are color-coded whether they contain (blue) or lack (red) subtelomeric sequences.

Minor Point

=====

1) It appears that several authors on the manuscript are employees of Oxford Nanopore. While I do not think their affiliation with Oxford Nanopore will affect the integrity of the study, this should be fully disclosed within the text. However, I do not see a “Conflict of interest” statement in the submitted manuscript.

Response: We have pointed out the affiliations in the “Competing Interest Statement”.

2) There seems to be significant variation in the number of reads observed in each of the libraries (e.g. ~3-5k reads for iPSCs in Supplementary Table 4, ~15-50k reads for fibroblasts in Supplementary Table 3 even though the authors showed that these fibroblasts have shorter telomeres than iPSCs, and ~20-30k reads for IMR90 in Supplementary Table 2.). What is the cause of this variation, and does it have an impact on the telomere length measurements by Telo-seq?

Response: We agree with the reviewer’s observation that there is a difference in the amount of total telomeric reads per flow cell in different samples. We do not have one single comprehensive explanation why the output for some samples is higher than for others. Potential factors that may contribute to the observed difference could be the cell sampling at different facilities, the number of cells per sample, and more efficient library preparation for some samples, among others.

The total number of telomeric reads per sample affects the level of downstream analysis. To evaluate how many telomeric reads are required to determine bulk telomere length, we performed modeling based on the HG002 telomere length measurements (Supplementary Table 1, Extended Data Fig. 1h). Our analysis suggests that bulk telomere length can be reliably determined with 500-1000 telomeric reads.

For further downstream analysis, like chromosome arm-specific telomere length, more reads are required than for bulk telomere length. The exact number of reads depends on the specific downstream analysis and how well the sample’s telomeric reads are aligning to the reference genome. For our analysis, we have included only chromosome arms that are supported by either 20 reads for the comparison between matched iPSCs and fibroblasts (Extended Data Fig. 6e-f), or 10 reads for the subtelomeric methylation analysis (Extended Data Fig. 8-9).

3) Can the authors perform a densitometric analysis on their TRF result in Figure 1b and compare the distribution to that observed with Telo-seq in Figure 1d (akin to Figure 2B in PMID: 34702734)?

Response: We have quantified the HG002 telomere length distribution based on TRF using TeloTool¹⁸ and WALTER¹⁹. The comparison between TRF and Telo-seq for HG002 is now shown in Extended Data Fig. 1f.

4) Lines 205-207 – This sentence can be written slightly clearer to emphasize the earlier study on heterogeneous telomere length in ALT cells. My initial impression was that this was an entirely new discovery from the authors’ work.

Response: To emphasize the known phenotype of ALT⁺ cells having heterogeneous telomere length, we modify the sentence to: “TRF analysis revealed that bulk telomere length in TERT⁺ cell lines was more tightly distributed, whereas, consistent with previous literature the telomeres of ALT⁺ cancer cell lines, were more heterogenous in length (Fig. 4a)”

5) The acrocentric arms (13p,14p,15p,21p,22p) are known to undergo frequent recombination (<https://www.nature.com/articles/s41586-023-05976-y>). It may therefore be difficult to distinguish telomeres that originate from each of these arms. This should be made clear in the figures and text.

Response: In the revised manuscript, we now discuss the challenges of chromosome arm-specific telomere length measurement, particularly acrocentric arms in the discussions and cite the

reference. Further, we have performed mapping simulations based on HG002 and CHM13 T2T genomes that are now presented in Extended Data Fig. 2c.

6) Line 363-364: Information on the bonito model used for calling telomere repeats is missing (e.g. version of the model, where to get it, how it was trained/obtained etc.)

Response: We have expanded the methods section including additional information about the customized Bonito telomere model. The Bonito model can be accessed through the GitHub repository.

7) I note that the GridION nanopore sequencer was used in the author's study. Given the wider accessibility of the P2/P2-solo sequencer, readers may be interested in the performance of Telo-seq on a PromethION flow cell. It would therefore be helpful to include this information in the manuscript if the authors have it on hand, though this is not absolutely necessary.

Response: In our study, we have intentionally used the GridION sequencer and the smaller MinION flowcells. Our rationale was that the reads obtained from a MinION flow cell are sufficient for our telomere analysis. Nevertheless, given that the major difference between MinION and PromethION flow cells is the number of pores, we expect the developed protocol to be equally compatible with a ONT sequencers using PromethION flow cells.

8) Extended Figure 1d – WGS and Telo-seq was performed on the same sample. Do both approaches give rise to similar telomere length measurements?

Response: We have repeated the method comparison experiment and provide the updated number of telomeric reads per flow cell and the bulk telomere length distribution in Extended Data Fig. 1d and e, respectively. The median telomere lengths obtained by WGS (3343 bp) and Telo-seq (3671 bp) are similar, whereas the AluI/MboI restriction enzyme digestion library showed a shorter median telomere length (1361 bp). Relative to Telo-seq, both WGS and restriction enzyme libraries have an accumulation of shorter telomeres below 1 kb. This is likely a consequence of not capturing the telomeric ends and therefore being more sensitive to either spontaneous or restriction enzyme-induced breaks inside the telomeric tracks. Differences between the Telo-seq telomere length distributions presented in Extended Data Fig. 1e and Fig. 1c result from the usage of different HG002 passages in both experiments.

9) Figure 2e – This is quite interesting. A nice and almost linear correlation of mean telomere length and population doubling was observed using Telo-seq.

Response: We thank the referee for their supportive remarks!

10) Line 371 – A minimum length of 500 bp of unique subtelomeric sequences was required for the reads to be considered. Is 500 bp sufficient to uniquely distinguish one chromosomal arm from another?

Response: Telo-seq captures the 5' end of the telomeric C-strand and the DNA is subsequently sequenced from the telomeric end in the direction of centromere. A random break at any point between the sequencing adapter at the telomeric terminus and the subtelomere can potentially lead to incorrect telomere length measurements. To prevent incorrect telomere length assignment and remove reads with random breaks, Telo-seq only takes into consideration the reads that contain terminal telomeric sequences and adjacent non-telomeric sequence. The minimum length

of non-telomeric sequence was chosen based on simulated EcoRV restriction enzyme digest of HG002 and CHM13. We have updated the minimal length to 60 bp in the revised manuscript to include all chromosome arms of the HG002 and CHM13 reference genome after the EcoRV-digested subtelomeres.

The minimal 60 bp unique/non-telomeric sequence length is frequently not sufficient to distinguish specific chromosome arms. To clarify the rationale for the usage of the minimal non-telomeric length filter, we have updated and expanded the methods section in the revised manuscript.

11) Line 517, 528, 543, 522 – Fig S4-S7 should be labelled as Extended Data Fig.

Response: We have changed the labeling according to the reviewer's recommendation.

12) Supplementary Tables – Does “No. of reads” represents the total number of reads in the library, the number of telomeric reads, or the number of filtered telomeric reads (based on criteria in line 366-371)? Can information on the total number of reads in each library be provided? Are the mean, std dev, median, IQR, values etc. calculated based on the length of telomeric repeats, or the length of the whole read?

Response: We have updated the Supplementary Tables according to the reviewer's recommendation. We have clarified the labeling and added the recommended additional information as highlighted by the reviewer.

Reviewer #3 (Remarks to the Author):

In this manuscript, “High resolution long-read telomere sequencing reveals dynamic mechanisms in aging and cancer,” Schmidt, Tyer, and Rugh et al., present their new method, “Telo-seq,” which is built to resolve bulk, chromosome arm-specific and allele-specific human telomere lengths using Oxford Nanopore Technologies’ native long-read sequencing. Telo-seq is a new method that uses telorette-based telomere adapters that are annealed to the G-overhang and ligated to the C-strand, digested with the blunt end restriction enzyme EcoRV, and includes a step to reduce concatemer ligation (dA-tailing). This method was shown to resolve telomere shortening in five population-doubling increments and revealed intrasample, chromosome arm-specific, allele-specific telomere length heterogeneity. Telo-seq showed some ability to reliably discriminate between telomerase- and ALT-positive cancer cell lines.

They compared their Telo-seq bulk TL measurements of two independent human cell line replicates to TRF results, and they were in line. Both also showed very similar subtelomere and telomere length distributions. They conclude that Telo-seq can reproducibly measure bulk and chromosome arm-specific TL dynamics in human cells, and resolve telomere shortening rates.

In human donor-derived fibroblasts, telomeres shortened with increasing age. Further, some chromosome arms consistently had shorter or longer telomeres, relative to the mean telomere length, raising the possibility that conserved chromosome arm-specific features influence telomere length, which are currently unknown. Telo-seq also allows analysis of higher resolution allele-specific telomere length.

Response: We thank the Referee for their support and constructive comments, which are making this a better manuscript. Please find detailed answers to the individual points of criticism below.

Thus, Telo-seq is a novel tool to study telomere biology during development, aging, and cancer at unprecedented resolution, and will be useful for the field. While the paper is generally sound, a few suggestions remain that would benefit from clarification:

1. They mention the A-tailed step to reduce concatemers, but don’t show much data on this, and it would be good to make sure there are not artifacts induced from the protocol.

Response: A-tailing is the recommended step in both the ONT and PacBio standard long-read library preparation protocols, aimed at preventing concatemers during library preparation. In our investigation into concatenated or chimeric reads, we found that approximately 0.105% (64 out of 61235 reads with telomeric motifs) of HG002 telomeric motif-containing reads were chimeric. Those reads are usually removed by the read structure filter.

2. Related to #1, they mention using a customized Bonito telomere model, but previous studies have shown that each version of the ONT software can introduce entirely new repeat calls and false positives, and it would be good to see a comparison (and ideally replication) of their results on a more recent version of the Bonito software.

Response: We appreciate the reviewer's insightful comment regarding the potential impact of different software versions on repeat calls and false positives. The customized Bonito telomere model was specifically designed for the R9.4.1 pore, with a primary focus on enhancing accuracy

within the telomere region. Our aim wasn't to develop a universal model applicable across different software versions.

It's worth noting that a general model for the R10 pore, offering enhanced accuracy for telomere analysis, has been released by ONT.

3. Their adapters used the canonical telomeric repeats and got a 75-fold increase in enrichment, but would a mixture of some of the known, non-canonical sequences included in the adapters help yield? Did the authors try this?

Response: We thank the reviewer for the suggestion. The reviewer is correct that we used only adapters with the canonical telomeric motif. We have not tested adapters with non-canonical sequences. Our reasons to use adapters that contain exclusively telomeric motifs are the following:

- 1) Previous literature, including long-read sequencing using PacBio HiFi sequencing⁷, revealed that telomeric variant repeats are more abundant closer to the telomere-subtelomere boundary and extremely infrequent at the telomeric terminus.
- 2) Well-established telomere length measurements methods like STELA¹ and TESLA² use only canonical telomeric motifs in their telomere adapters.
- 3) The selection for canonical telomeric sequence is highest at the very end, and such true telomeric motifs have been shown to be required for efficient end protection and t-loop formation.

4. Related to #3, did the authors observe any non-canonical telomere variants? These have been documented in the literature and would be expected here as well.

Response: In line with previous reports^{7, 20-23}, we have also detected telomere variant repeats. These telomere variant repeats increase towards the telomere-subtelomere boundary^{7, 23}. In this manuscript, we have focused on the human telomere length assessment in various cell types and conditions. In the future, we think that long-read sequencing of telomeres and Telo-seq will provide valuable insights in the distribution and localization of telomere variant repeats. However, for this manuscript, we consider an in-depth evaluation of telomere variant repeats to be beyond the scope.

5. They showed a general trend of shorter telomeres with age, but there was a 82-year-old donor that was an outlier; was there anything clinically or health-wise unique about this individual that could explain this difference?

Response: We appreciate the reviewer's interest in the 82-year-old individual. However, we do not have any further information about this individual or the other individuals in the cohort, due to privacy concerns and restrictions. We like to highlight previous work that showed that the telomere length in different individuals of a population can vary dramatically, even at the same age⁶⁻⁸. Considering these studies, we think that the 82-year-old individual is a good example for the telomere length heterogeneity present in the human population; therefore, we postulate that the 82-year-old individual may be an individual that belongs to the subgroup of the population that have longer telomeres relative to their respective age group.

6. The authors describe the methylation patterns in the CpG islands from their data; but did they look at the prediction of other base modifications? This might be interesting to explore, in this, or a follow-up paper.

Response: We thank the reviewer for the comment on and the interest in our CpG islands analysis. We agree with the reviewer that base modifications are an exciting additional piece of information that can be harvested by native ONT long-read sequencing. In the revised manuscript, we have included a summary heatmap showing the mean methylation status close to the telomere per sample and chromosome arm (Extended Data Fig. 7). We have not investigated additional base modifications, mainly to the lack of a trained model for the used R9.4.1 chemistry. We agree with the reviewer that, with the R10 chemistry updates and improved base modification calling, a detailed analysis of base modifications at telomeres would be a very interesting future avenue to explore in a follow-up paper, especially given that more base modifications will be made available in future.

7. The software referred to in this paper should be made available.

Response: We thank the reviewer for this comment. At the initial submission we have included the link to the GitHub repository only in the code and software submission checklist and reporting summary file, as instructed by Nature Communications. In the revised manuscript, we have added a Code Availability section after the methods.

8. The raw data used for the paper is not available, making it impossible to validate the findings, and this too needs to be made available.

Response: We thank the reviewer for indicating that the data availability statement in the initially submitted manuscript was missing. We have deposited the raw data at <https://www.ncbi.nlm.nih.gov/bioproject/PRJNA1040425>. At the initial submission, the link to raw data was only listed in the reporting summary file as per instructions from the journal. In the revised manuscript, we have added the information in the Data Availability section after the methods.

9. They conclude that, upon induced pluripotency, shorter telomeres are preferentially elongated by telomerase; however, the overall order of chromosome arm-specific telomere length remained conserved between fibroblasts and iPSCs. This should be expanded upon in the discussion or more in the results.

Response: We have followed the reviewer's advice and have expanded this point in the discussion of the revised manuscript.

10. They propose that Telo-seq is an effective method to evaluate the TMM of cancer cells, as it recapitulates the differences in TL distributions in telomerase vs ALT+ cells. How well would this expand to other cell lines and genomes?

Response: We detect the described differences in 5 ALT⁺ and 5 TERT⁺ cancer cell lines. Furthermore, the telomere maintenance mechanism-dependent telomere length distribution differences have been previously reported²⁴. Therefore, we would expect that the results would be reproducible with other cell lines.

11. Importance/significance: This study highlights the potential of human telomere long-read sequencing and sets the stage to investigate human telomere dynamics in unprecedented detail during development, aging and disease. This could be discussed more.

Response: We thank the reviewer for the suggestion and have expanded the outlook on telomeropathies.

12. They have referenced some prior work in telomere sequencing, including data from the Telomere-to-Telomere (T2T), and Genome in the Bottle Consortiums, but there are other papers which have already used ONT and long-read data to map telomere length and variation, which should be cited, including: <https://pubmed.ncbi.nlm.nih.gov/33242406/> <https://pubmed.ncbi.nlm.nih.gov/33242411/>

Response: We thank the reviewer for the suggestion. In the introduction of the revised manuscript, we have now included Luxton et al., 2020²⁵. Further, we added Smoom et al. 2023⁹, who use ONT sequencing to investigate telomeres in mouse and compare them to human cells.

References:

1. Baird, D.M., Rowson, J., Wynford-Thomas, D. & Kipling, D. Extensive allelic variation and ultrashort telomeres in senescent human cells. *Nat Genet* **33**, 203-207 (2003).
2. Lai, T.P. et al. A method for measuring the distribution of the shortest telomeres in cells and tissues. *Nat Commun* **8**, 1356 (2017).
3. Hastie, N.D. et al. Telomere reduction in human colorectal carcinoma and with ageing. *Nature* **346**, 866-868 (1990).
4. Vaziri, H. et al. Loss of telomeric DNA during aging of normal and trisomy 21 human lymphocytes. *Am J Hum Genet* **52**, 661-667 (1993).
5. Slagboom, P.E., Droog, S. & Boomsma, D.I. Genetic determination of telomere size in humans: a twin study of three age groups. *Am J Hum Genet* **55**, 876-882 (1994).
6. Aubert, G., Baerlocher, G.M., Vulto, I., Poon, S.S. & Lansdorp, P.M. Collapse of telomere homeostasis in hematopoietic cells caused by heterozygous mutations in telomerase genes. *PLoS Genet* **8**, e1002696 (2012).
7. Tham, C.Y. et al. High-throughput telomere length measurement at nucleotide resolution using the PacBio high fidelity sequencing platform. *Nat Commun* **14**, 281 (2023).
8. Alder, J.K. et al. Diagnostic utility of telomere length testing in a hospital-based setting. *Proc Natl Acad Sci U S A* **115**, E2358-E2365 (2018).
9. Smoom, R. et al. Telomouse—a mouse model with human-length telomeres generated by a single amino acid change in RTEL1. *Nat Commun* **14**, 6708 (2023).
10. Stein, G.H. SV40-transformed human fibroblasts: evidence for cellular aging in pre-crisis cells. *J Cell Physiol* **125**, 36-44 (1985).
11. Wright, W.E., Pereira-Smith, O.M. & Shay, J.W. Reversible cellular senescence: implications for immortalization of normal human diploid fibroblasts. *Mol Cell Biol* **9**, 3088-3092 (1989).
12. Shay, J.W., Pereira-Smith, O.M. & Wright, W.E. A role for both RB and p53 in the regulation of human cellular senescence. *Exp Cell Res* **196**, 33-39 (1991).
13. Nassour, J. et al. Telomere-to-mitochondria signalling by ZBP1 mediates replicative crisis. *Nature* **614**, 767-773 (2023).
14. Nassour, J. et al. Autophagic cell death restricts chromosomal instability during replicative crisis. *Nature* **565**, 659-663 (2019).
15. Rufer, N., Dragowska, W., Thornbury, G., Roosnek, E. & Lansdorp, P.M. Telomere length dynamics in human lymphocyte subpopulations measured by flow cytometry. *Nat Biotechnol* **16**, 743-747 (1998).
16. Martens, U.M. et al. Short telomeres on human chromosome 17p. *Nat Genet* **18**, 76-80 (1998).
17. Graakjaer, J. et al. The pattern of chromosome-specific variations in telomere length in humans is determined by inherited, telomere-near factors and is maintained throughout life. *Mech Ageing Dev* **124**, 629-640 (2003).
18. Gohring, J., Fulcher, N., Jacak, J. & Riha, K. TeloTool: a new tool for telomere length measurement from terminal restriction fragment analysis with improved probe intensity correction. *Nucleic Acids Res* **42**, e21 (2014).
19. Lycka, M. et al. WALTER: an easy way to online evaluate telomere lengths from terminal restriction fragment analysis. *BMC Bioinformatics* **22**, 145 (2021).
20. Conomos, D. et al. Variant repeats are interspersed throughout the telomeres and recruit nuclear receptors in ALT cells. *J Cell Biol* **199**, 893-906 (2012).
21. Lee, M. et al. Telomere extension by telomerase and ALT generates variant repeats by mechanistically distinct processes. *Nucleic Acids Res* **42**, 1733-1746 (2014).
22. Lee, M. et al. Telomere sequence content can be used to determine ALT activity in tumours. *Nucleic Acids Res* **46**, 4903-4918 (2018).

23. Grigorev, K. et al. Haplotype diversity and sequence heterogeneity of human telomeres. *Genome Res* **31**, 1269-1279 (2021).
24. Bryan, T.M., Englezou, A., Gupta, J., Bacchetti, S. & Reddel, R.R. Telomere elongation in immortal human cells without detectable telomerase activity. *EMBO J* **14**, 4240-4248 (1995).
25. Luxton, J.J. et al. Temporal Telomere and DNA Damage Responses in the Space Radiation Environment. *Cell Rep* **33**, 108435 (2020).

REVIEWERS' COMMENTS

Reviewer #1 (Remarks to the Author):

see attachment

Reviewer #1 (Remarks to the Author):

Tobias T et al have demonstrated the capability to measure telomere length using Oxford Nanopore long-read sequencing platform. The authors also design a method (Telo-seq) to enrich for telomere containing genomic DNA, which resulted in more than 75-fold increase in telomeric read output comparing to whole genome sequencing. A customized Bonito telomere model was developed for basecalling and sequencing data analysis, which should be useful for scientists studying telomere length in cancer and aging. Allele-specific telomere length can be measured for all 44 chromosomal ends excluding the sex-determinant chromosomes, showing potential chromosome arm-specific length heterogeneity. In addition to these findings, the authors also apply this new telomere sequencing method to successfully differentiate TERT+ cancer cells and ALT+ cancer cells by measuring the dispersion of their telomere length distribution.

Overall, the results are clear, and the method is interesting. However, the current manuscript is missing key details for comprehensive evaluation of the data, which may limit this new approach to recover accurate mean telomere lengths as well as allele-specific telomere lengths.

Response: We thank the Referee for their support and constructive comments, which are making this a better manuscript. Please find detailed answers to the individual points of criticism below.

Major comments:

1. The Telo-seq method adapted a similar strategy as STELA by ligating telomere-based telomere adapters to the C strand of telomeric ends. This will inadvertently result in loss of information for telomere G tail overhangs, which is known to vary from 100-300 nucleotides in length in human cells. Therefore, the newly established Telo-seq may result in underestimation of actual telomere length. Given the peripheral blood mononuclear cells telomere shortening rate is about 40-50 nts per year, whether this method can be utilized for accurate measurement of telomere length shortening rate in general population remains to be validated.

Response: We agree with the reviewer that Telo-seq uses like STELA¹ and TESLA² telomere-based telomere adapters and that Telo-seq is therefore not providing any information about the telomeric G-rich overhang length. The same argument is true for STELA and TESLA, which are both well-established methods in the telomere field to measure telomere length. Thus, if the telomeric G-rich overhang length is critical for a specific research question, alternative methods to STELA, TESLA and Telo-seq have to be applied.

The telomere shortening rate of ~30-50 nt in adult peripheral blood mononuclear cells have been determined based on a larger cohort of individuals. First, the individual's telomere length was determined with TRF³⁻⁵ or flow FISH⁶, for example. Second, the determined telomere length was plotted against the donor age and linear regression analysis performed. Our data on IMR90^{E6E7} fibroblasts at different population doublings suggest that Telo-seq can differentiate telomere length in samples 5 population doublings apart (Fig. 2). Given that Telo-seq has similar or higher resolution than TRF and age-dependent Telo-seq-based telomere length decrease is also observed in the fibroblasts of the investigated aging cohort (Fig. 4c), we therefore postulate that, comparable to previous work, an averaged peripheral blood mononuclear cell telomere shortening rate per year from a cohort can be determined with Telo-seq.

Comment: Based on previous estimation: in healthy human cells, telomeres are typically 3 to 15 kilobase pairs (kb) in length. They shorten at rates of 50–200 base pairs per replication,

and undergo 30–60 population doublings before becoming senescent⁷. While the authors have indicated that “Telo-seq can differentiate telomere length in samples 5 population doublings apart (Fig. 2)”, the difference in telomere length between samples that are 5 population doubling apart is about 200 nt based on the authors’ own estimation of 39 bp per doubling, which is equivalent to 5 year worth of telomere shortening in human peripheral blood mononuclear cells. While I have no doubt that the current Telo-seq has achieved higher resolution than TRF in bulk telomere length measurement, I am wondering how the authors have the confidence to differentiate the annual telomere shortening rate that is merely ~30-50 nt in adult peripheral blood mononuclear cells, which is about the shortening rate of one doubling in IMR90^{E6E7} cells.

Further, as shown in Supplementary Table 4, the number of reads in about 40% (107 out of 264) alleles is below 250. The telomere length estimated in these alleles will have much larger standard error (>100bp) as indicated in the simulation study in a recent paper measuring telomere length using Nanopore long-read sequencing platform⁸. Therefore, the standard error is much larger than the estimated telomere shortening rate by the authors, which is about 39bp per doubling. Hence, it is impossible to get accurate allele-specific telomere shortening rate using this set of data considering the limited signal to noise ratio (39bp to 100bp in 40% allele groups).

The lack of ability to accurately measure the G rich overhang brings more doubt for the accuracy of current method. The lack of accuracy for allele-specific telomere length using STELA and TESLA was well known.

By the way, results from Sanchez, S.E. et al⁸ and Karimian, K. et al⁹ has adopted a different ligation method to enable the sequence of the G tail overhangs using Nanopore long-read sequencing platform. Whether there is difference in the accuracy of telomere length measurement using these two different methods remains to be addressed.

2. The customized Bonito telomere model that was developed for basecalling in Oxford Nanopore sequencing data analysis will be valuable to the research interest of telomere length in cancer and aging. Please explain the detailed training process for this new model. Please also describe the detailed normalization steps that have been applied to read coverage.

Response: We thank the reviewer for highlighting this point. We have expanded the method section in the revised manuscript.

3. Please show the raw read coverage in all figures, which is essential to judge the accuracy of telomere length measurement.

Response: We appreciate the reviewer’s comment. In the revised manuscript, we have added the number of telomeric reads corresponding to each bulk telomere length analysis figure panel. A summary for each flow cell and sample including number of telomeric reads is shown in the Supplementary Table 1, 3, 7, 8, 10, 11. Due to figure font size limitations we have added additional Supplementary Tables 2, 4 5, 7, and 9 to the revised manuscript including detailed information about chromosome arm-specific telomere length analysis, including number of telomeric reads per chromosome arm. The revised figure legends link to the corresponding Supplementary tables.

Comment: If the font size limitation is the concern, the read depth can be included in the legend.

4. The telomere length estimated from HG002 Chr. 21p before haplotype phasing is above 9kb (Fig. 1e and lines 107-108). However, the lengths were estimated to be 4-5kb and 10-11kb for maternal and paternal alleles separately (Extended Data Fig. 4a). So, if the two alleles

have been sequenced in similar depth, the telomere length of Chr. 21p should be much smaller than 9kb. Chr.1p has the same problem. Could you elaborate more regarding these differences. In addition, the allele-specific telomere lengths for paternal Chr. 13p and Chr. 22p are missing in Extended Data Fig. 4a as well.

Response: The reviewer's assumption that the differences between the chromosome arm-specific telomere length of HG002 and the two corresponding alleles can be due to different allelic coverages (Supplementary Table 2 and 5) is correct. This is likely the reason why the 1p chromosome arm-specific telomere length is closer to the maternal allele-specific telomere length (610 maternal and 337 paternal reads) (Supplementary Table 2 and 5). For Chr. 21p, both alleles have roughly the same number of reads (133 maternal and 113 paternal). However, we like to note that the boxplots show the median telomere length for each chromosome arm. Therefore, in the cases, in which the maternal and paternal telomere length are substantially different from each other and form distinct telomere length populations, even with same allelic coverage, the median will be closer to one allelic population than to the other one and likely be different than the mean telomere length of both populations. As suggested by the reviewer in major comment 3, we have added the underlying number of reads to Supplementary Table in the revised manuscript.

The paternal HG002 alleles of chromosome arm 13p and 22p are not shown in Fig. 3, because the mapping quality of both paternal alleles are below the applied mapping quality filter of 5. We have added a comment in the figure legend to explain why only the telomere length of the maternal 13p and 22p alleles are shown. We have also added a plot of the mapping quality per allele in HG002 in Extended Data Fig. 4a of the revised manuscript.

Comment: One of the major claims of the paper is "allele-specific human telomere lengths using Oxford Nanopore Technologies' native long-read sequencing." If the method failed to provide the allele-specific length for all chromosomal ends, then the authors should not make the claim.

In addition, no information is provided for the X and Y chromosomal ends for HG002.

5. In Extended Fig. 4a, it is very clear that both paternal and maternal allele in HG002 chr18q have two sets of reads in it: One is about 4-6kb, the other is about 8-10kb. Why the two alleles in this specific arm exhibit 2 sets of telomeres with different length?

Response: We agree with the reviewer that for both HG002 chromosome 18q alleles the majority of reads cluster around 8-10 kb and there is an accumulation of reads at 4-6 kb. The appearance of shorter telomeres could be indicative of a subpopulation of cells that have acquired shorter telomeres at chromosome 18q, for example due to rapid telomere shortening. The relatively long telomere length of 4-6 kb will likely not interfere with growth as other HG002 chromosome arms have shorter telomere length.

Comment: Any evidence to support the claim that "The appearance of shorter telomeres could be indicative of a subpopulation of cells that have acquired shorter telomeres at chromosome 18q, for example due to rapid telomere shortening."

Alternative explanation is the two alleles from 18q have different telomere length, and the difference in abundance is due to difference in read depth of the two alleles. Same problem throughout the study.

6. It is known that telomere length is reversely correlated with age. However, the authors have identified an 82-year-old individual who has abnormal longer telomere compared to the other individuals in his/her age group (Fig. 3a). How many replicates have been sequenced using Oxford Nanopore platform for each individual? Have the authors compared using the

traditional TRF method to measure the telomere length in these eight individuals to validate the accuracy of their telomere length estimation using Oxford Nanopore platform.

Response: The reviewer stresses an important point with this comment. Previous work has shown that the telomere length of individuals in the population even in the same age group is very heterogeneous ^{6, 10, 11}. Whereas all of these studies observe a reverse correlation of telomere length with age, there are also always individuals reported that have abnormal long telomeres relative to their respective age group, but also in comparison to individuals several decades younger. Therefore, we think that the 82-year-old individual present in our small aging cohort is indeed one of these individuals, who has relatively long telomeres for their age. As we have shown reproducibility of Telo-seq results for HG002 and cancer samples (Supplementary Table 1, 11), we have sequenced one replicate per sample for the aging cohort samples (Supplementary Table 6). We have compared TRF and Telo-seq telomere length for the HG002, IMR90 ^{E6E7} and cancer samples and got overall consistent results between both methods (Fig. 1, 2, 5). Due to limited access to sample, we could not run a TRF on the eight individuals in the aging cohort.

Comment: Additional replicate or data from alternative method is necessary to confirm the accuracy of the telomere length measurement using the Telo-seq.

7. While Telo-seq may be able to distinguish TERT+ cancer cells and ALT+ cancer cells, the telomere length distribution in ALT cells obtained from Nanopore sequencing seem to be much shorter than the traditional teloblot (compare Fig. 4b and Fig. 4a). What is the potential cause for such discrepancy? The intense high molecular weight fractions in ALT samples seem to indicate incomplete genomic DNA digestion as well.

Response: We agree with the reviewer that the telomere length distribution of the ALT+ cancer cells seem to be longer in the TRF analysis than in Telo-seq.

Based on the methodical differences there are several explanations:

- 1) For the TRF analysis, high-molecular weight genomic DNA is digested with restriction enzymes (Alu/Mbol in our protocol) that frequently cut outside the telomeric motif. So telomeric TRF fragments contain telomeric repeats and adjacent subtelomeric sequence. Therefore, the detected telomere length in TRF is the sum of the telomere and the remaining subtelomere present in the fragment, whereas for Telo-seq we are only plotting the telomere length. Longer estimated telomere length based on TRF relative to telomere length determined by long-read sequencing have been also reported in a recent study for mouse cells ¹².

Comment: If this is the case, it is certainly possible to estimate the telomere length based on the Alu/Mbol sites that can be identified in the long-read sequencing data from Nanopore. Please provide the data.

- 2) TRF uses a telomeric probe to stain telomeric repeat containing fragments. As consequence, fragments with more telomeric repeats (longer telomeres) will allow more probe binding, resulting in higher intensity for longer than for shorter telomeres. For example, HT-29 and SK-N-AS have more intense telomere distribution than Calu-3 (Fig. 5a). Thus, shorter telomeres appear weaker on a TRF than longer telomeres. Contrary, in Telo-seq plots every telomeric read will be equally visualized, independently of its length.

Comment: This is certainly more reasonable. But it does not explain the drastic difference in telomere length shown in TRF and Telo-seq.

- 3) Fragments in a TRF analysis are separated on an agarose gel which runs non-linear. Smaller fragments will be separated on a larger distance. Consequently, longer fragments run closer to each other and are relatively compressed compared to shorter fragments. Thus, in addition to the probe binding, the gel running behavior is contributing to the visual impression of higher fragments being more abundant than shorter fragments, especially in samples with very heterogenous telomere length distributions like ALT⁺ cancer cells.

Comment: Large fragment will also have less efficiency to transfer as well.

- 4) In TRF, the probe is staining any fragment which contains telomeric repeats, including interstitial telomeric repeats. These interstitial telomeric repeats can be generated in a chromosome fusion event and will lead to distinct bands in a TRF, for example. In contrast, Telo-seq captures the telomeric ends by using telomere-based adapters specific for the telomeric overhang and is therefore very unlikely to sequence these interstitial telomeric repeats. In highly rearranged and genetically unstable genomes like cancer genomes, especially ALT⁺ cancer cells, these interstitial telomeric repeats may be more common.

Comment: Any actual evidence to support such assumption?

Taken together, we like to argue that these methodological differences contribute to and explain the impression that ALT cells seem to have longer telomeres based on TRF analysis.

Incomplete digestion of genomic DNA could be an explanation for high molecular weight fraction. However, based on the SybrGold staining that we have done for the gel prior to Southern blotting (Reviewer Fig. 1), we do not think that the high molecular weight fragments are due to incomplete digestion.

Comment: Looking at the SybrGold staining, it appears to have high molecular weight fragments at the very top of each lane on the gel. Please provide SybrGold staining gel picture with increased contrast.

Reviewing Fig. 1. TRF analysis of cancer cell lines. AluI/MboI digested genomic DNA of 5 TERT⁺ and ALT⁺ cancer cell lines were separated on a 0.7% agarose gel. Left) agarose gel stained with Sybr Gold (1:10,000 in 0.5x TBE, 40 min, RT). Right) TRF using TelG telomeric probe (as shown in Fig. 5a)

8. The author indicated that “In ALT+ cells, we could measure telomeres from 47 bp to 134.7 kb in length.” The 47 bp telomere seem to be too short. How often does the sample show telomere reads with such short length (percentage)?

It is highly possible that such fragments (47 bp) may be resulted from random shearing of genomic DNA during genomic DNA extraction.

Response: The very short telomere lengths, below 50 and 100 bp, are generally very infrequent (Reviewer Fig. 2). Only the ALT⁺ cell line SK-LU-I with the shortest telomere length distribution had a significant accumulation of very short telomeres below 100 bp (~4%) (Reviewer Fig. 2 c). Telomeres between 1-50 bp are very infrequent (Reviewer Fig. 2d).

We cannot completely rule out that some of the reads with very short telomeres are due to random sharing during DNA prep or handling. However, as we are using a probe against the overhang to introduce the sequencing adapter, we consider it as rather unlikely that the randomly shared DNA will produce an overhang that is compatible with the probe. Further, as postulated for cells without an active telomere maintenance mechanism, short telomeres (<1 kb) are accumulating in IMR90^{E6E7} with increasing population doubling (Fig. 2d, Reviewer Fig. 2), suggesting that the short telomeres are present in these samples and not (exclusively) a product of sample preparation (Reviewer Fig. 2). Finally, independently to us, the Artandi lab also reports the presence of short telomeres in their preprint on human telomere nanopore sequencing (<https://doi.org/10.1101/2023.11.29.569263>).

Reviewer Fig. 2. Distribution of telomere length in cancer and IMR90^{E6E7}. **a**, binned telomere length in base pairs (bp) of cancer cell lines. **b**, binned telomere length IMR90^{E6E7} at indicated population doublings (PD). **c**, bar graph of the percentage of telomeres below 100 bp. Number of telomeric reads with a telomere length between 1-100 bp is shown above the bar. **d**, bar graph percentage of telomeres below 50 bp. Number of telomeric reads with a telomere length between 1-50 bp is shown above the bar.

Comment: Based on Reviewer Fig. 2, telomeres with 1-100 bp exist even in IMR90^{E6E7} that is in earlier passage of 66.2. Such results will indicate substantial induction of telomere dysfunctional foci in IMR90^{E6E7} that is in earlier passage. Has the author done immunohistochemistry to confirm such finding? I can foresee chromosomal ends with extremely short telomere do exist. However, one will expect such extremely short telomeres will increase with passage in vitro. Based on Reviewer Fig. 2 above, the fraction of telomeres with 1-100 bp seem to be constant during in vitro passage. One will expect such fraction increases with in vitro passages as the telomeres that are 101-500 bp in length that is shown in above Fig. 2b. Can the authors provide explanation for such discrepancy for telomeres that are 1-100 bp and 101-500 bp?

9. The Allele-specific telomere length measurement is certainly one of the most interesting results in this manuscript.

Response: We thank the reviewer for this comment that encouraged us to move the mapped phased HG002 telomere length (submitted manuscript Extended Data Fig. 4a) as independent main text Figure 3 in the revised manuscript. We moved the paragraph about allele specific telomere length assignment up in the revised manuscript text right after the “Telo-seq resolves telomere shortening” paragraph.

While the authors showcase the capability to measure the telomere length of all 88 alleles, whether they are accurate remains a major question. For example: While the author highlighted their ability to accurately measure telomere shortening rate (average 39 bp per PD) in IMR90E6E7 cells in serial passage in vitro, the allele-specific telomere length shortening seems to have a lot of heterogeneity. In many instances (Fig. 2f, Chr. 22p PD106.1; Chr. 14p PD106.1; Chr 12q PD106.1, etc) the allele-specific telomere lengths in the late passage are much longer than IMR90E6E7 in earlier passages. These discrepancies of allele-specific telomere length bring a lot of doubt on the accuracy of allele-specific telomere length measurement using current method.

Response: As the reviewer has correctly pointed out, the median chromosome arm-specific telomere length is increasing in some chromosome arms especially in the later passages (PD106). However, we like to note that these cells (IMR90^{E6E7} PD106) are encountering replicative crisis¹³⁻¹⁵. Crisis, or mortality stage 2, is a telomere-dependent proliferation barrier, in which most cells die an autophagy-dependent, innate immunity-driven cell death^{16, 17}. Thus, cells with dysfunctional telomeres will be efficiently removed from the population. As telomere length in a cellular population is distributed around an averaged telomere length¹⁸, there will be some cells with overall shorter, or with short telomeres, at only some chromosome arms. We postulate that this subpopulation of cells will experience crisis earlier than the rest of the population. Thus, our interpretation of the results is that there is counterselection against the subpopulations of cells that have very short bulk telomeres, or very short telomeres, at individual chromosome arms in cells approaching crisis as they are experiencing crisis earlier and are removed from the population. Consequently, under these selection conditions, clones with longer telomeres will outgrow the ones with shorter ones. For the last sampling point (PD106), but likely also for cells that bypass the senescence plateau (PD96), there are two processes affecting the chromosome arm-specific telomere length measurements: telomere shortening and selection for a population of cells with telomeres long enough to protect chromosome ends. Consequently, we have excluded the two last timepoints for the calculation of chromosome arm-specific shortening rates (Extended Data Fig. 3c). In the revised manuscript we have included additional benchmarking figures based on HG002 Telo-seq analysis (Extended Data Fig. 1 and 2), updated the chromosome arm-specific telomere length figures using a mapping quality of 20 and discussed current limitations of chromosome arm-specific telomere length assignment in the "Discussion".

Comment: If the authors' speculation that "Consequently, under these selection conditions, clones with longer telomeres will outgrow the ones with shorter ones." is valid. One will expect this should be true for all the chromosomal ends at PD106.1. However, this is certainly not the case.

Similar arguments apply to many of the allele-specific telomere length estimation using reads that are less than 20 in total (Supplementary Table 4_Telo-seq). Are these estimations accurate? Since the authors did not provide additional data to rectify their conclusion, doubts remain on the accuracy of allele-specific telomere length measurement using Telo-seq at current state.

In addition, only the chromosomal end specific telomere lengths are presented, but not the allele-specific telomere lengths of IMR90E6E7 cells are shown for each PD? Given the differences of allele-specific telomere length (Extended Data Fig. 4c), plotting the allele-specific telomere shortening rate in Fig. 2f will be important.

Response: We agree with the reviewer that allele-specific shortening rates would be very interesting. However, resolving allele-specific telomere length is challenging and, given the heterogeneity, requires a high coverage per chromosome arm. Phasing quality also depends on the reference genome. We present a de novo approach (Extended Data Fig. 4b,c);

however, as discussed in the manuscript, this approach is not optimal, and we cannot resolve all alleles for every chromosome arm. One key issue is that for IMR90^{E6E7} as well as for all other samples in this study, except for HG002, there is no phased reference genome and parental genome information available. Given these challenges we do not feel confident in calculating allele-specific shortening rate at this point.

Comment: If this is the case, the authors should not claim to achieve “allele-specific human telomere lengths using Oxford Nanopore Technologies’ native long-read sequencing” in general.

10. By profiling allele-specific telomere length using donor-derived fibroblasts, the authors have indicated that some chromosome arms consistently had shorter or longer telomeres. They further proposed that there are potential conserved chromosome arm-specific features influence telomere length. Similar results have been reported previously (Londono-Vallejo J.A. NAR 2001; Martens U.M. Nature Gen. 1998; Graakjaer J. Mech. Ageing Dev. 2003 etc) using other telomere length measurement methods, such as FISH. However, there are no consistency on which chromosomal end has the shortest or the longest telomere.

Response: We thank the reviewer for this comment. Given the high heterogeneity of the telomere length on the chromosome arm level and the relatively small cohort size (≤ 20 individuals in previous^{19, 20} and 9 in our work), we believe that to comprehensively address the question whether specific telomeres at some chromosome arms are consistently shorter or longer requires a large, diverse, and well-controlled cohort. Nevertheless, despite some discrepancy with and within previous data, the five shortest (chromosome 21p, 19p, 19q, 16q, 9q) and longest telomeres (chromosome 18q, 12q, 3p, 4q, 5p) identified in our cohort are also detected to be shorter and longer in two previous studies^{19, 20}. We have expanded this point in the discussion of the revised manuscript.

Similar results were observed in Extended Data Fig. 4a: most of the two alleles from the same chromosomal end seems to have vast different telomere length. The allele-specific telomere length for the eight samples should be shown in Extended data Fig. 3c. In summary, whether there is conserved chromosome arm-specific features influence telomere length remains to be addressed.

Response: Telomere phasing without the sample’s phased reference genome is difficult and our de novo phasing approach frequently results in alleles and chromosome arms that cannot be resolved (Extended Data Fig. 4b,c). In addition, we do unfortunately not have enough coverage to present allele-specific telomere length for the eight fibroblasts.

We agree with the reviewer that it is an open question, whether conserved chromosome arm-specific features exist that influence human telomere length. Our analysis is limited to only 9 fibroblast samples. Whether there are chromosome arm-specific features that influence telomere length, and which chromosome arms have the longest/shortest telomeres has to be determined in future studies. To answer these questions comprehensively, analysis of a much larger, well-controlled, diverse cohort is required. This is beyond the scope of this manuscript.

11. What is the meaning of the methylation data? There is a lot of heterogeneity, which may be due to the sequencing method itself or read depth. Is the methylation results shown in each panel in extended data Fig. 6 average of multiple reads from the same allele? The results have shown that ALT+ cancer cell lines were observed to be hypomethylated compared to TERT+ cancer cell lines. However, it is also mentioned that not all the arms were equally hypomethylated. Can the authors specifically list out the hypomethylated/hypermethylated chromosome arms? And within the differentially methylated arms, what are the differentially methylated regions (DMR)? Does the ALT+ cell lines have similar DMR compared to TERT+

cell lines? Furthermore, it is also important to compare the methylation pattern of cancer cells with normal cells (HG002 and IMR90).

Response: The advantage of native nanopore sequencing and Telo-seq is that, in addition to the sequence information and telomere length, the methylation status of the CpG sites closest to the telomeres can be obtained. We agree there is inter- and intra-sample heterogeneity. However, to our knowledge this type of analysis has not been done in this detail before and highlights which type of questions can be answered by Telo-seq.

We thank the reviewer for suggesting a summary about the methylation status of the CpG sites closest to the telomeres in controls and cancer cell lines. This will provide a fast overview and facilitate the search for the chromosome arms where the methylation status differs from controls. We have generated a heatmap that shows the averaged methylation of every chromosome arm for the indicated sample and added it to Extended Data Fig. 7.

Comment: I would like to point out that there are two recent papers^{21,22} reporting the DNA methylation profiling at telomeres.

1. Baird, D.M., Rowson, J., Wynford-Thomas, D. & Kipling, D. Extensive allelic variation and ultrashort telomeres in senescent human cells. *Nat Genet* **33**, 203-207 (2003).
2. Lai, T.P. et al. A method for measuring the distribution of the shortest telomeres in cells and tissues. *Nature communications* **8**, 1356 (2017).
3. Hastie, N.D. et al. Telomere reduction in human colorectal carcinoma and with ageing. *Nature* **346**, 866-868 (1990).
4. Vaziri, H. et al. Loss of telomeric DNA during aging of normal and trisomy 21 human lymphocytes. *Am J Hum Genet* **52**, 661-667 (1993).
5. Slagboom, P.E., Droog, S. & Boomsma, D.I. Genetic determination of telomere size in humans: a twin study of three age groups. *Am J Hum Genet* **55**, 876-882 (1994).
6. Aubert, G., Baerlocher, G.M., Vulto, I., Poon, S.S. & Lansdorp, P.M. Collapse of telomere homeostasis in hematopoietic cells caused by heterozygous mutations in telomerase genes. *PLoS Genet* **8**, e1002696 (2012).
7. Harley, C.B., Futcher, A.B. & Greider, C.W. Telomeres shorten during ageing of human fibroblasts. *Nature* **345**, 458-460 (1990).
8. Sanchez, S.E. et al. Digital telomere measurement by long-read sequencing distinguishes healthy aging from disease. *bioRxiv* (2023).
9. Karimian, K. et al. Human telomere length is chromosome specific and conserved across individuals. *bioRxiv* (2024).
10. Tham, C.Y. et al. High-throughput telomere length measurement at nucleotide resolution using the PacBio high fidelity sequencing platform. *Nat Commun* **14**, 281 (2023).
11. Alder, J.K. et al. Diagnostic utility of telomere length testing in a hospital-based setting. *Proc Natl Acad Sci U S A* **115**, E2358-E2365 (2018).
12. Smoom, R. et al. Telomouse—a mouse model with human-length telomeres generated by a single amino acid change in RTEL1. *Nat Commun* **14**, 6708 (2023).
13. Stein, G.H. SV40-transformed human fibroblasts: evidence for cellular aging in pre-crisis cells. *J Cell Physiol* **125**, 36-44 (1985).
14. Wright, W.E., Pereira-Smith, O.M. & Shay, J.W. Reversible cellular senescence: implications for immortalization of normal human diploid fibroblasts. *Mol Cell Biol* **9**, 3088-3092 (1989).

15. Shay, J.W., Pereira-Smith, O.M. & Wright, W.E. A role for both RB and p53 in the regulation of human cellular senescence. *Exp Cell Res* **196**, 33-39 (1991).
16. Nassour, J. et al. Telomere-to-mitochondria signalling by ZBP1 mediates replicative crisis. *Nature* **614**, 767-773 (2023).
17. Nassour, J. et al. Autophagic cell death restricts chromosomal instability during replicative crisis. *Nature* **565**, 659-663 (2019).
18. Rufer, N., Dragowska, W., Thornbury, G., Roosnek, E. & Lansdorp, P.M. Telomere length dynamics in human lymphocyte subpopulations measured by flow cytometry. *Nat Biotechnol* **16**, 743-747 (1998).
19. Martens, U.M. et al. Short telomeres on human chromosome 17p. *Nat Genet* **18**, 76-80 (1998).
20. Graakjaer, J. et al. The pattern of chromosome-specific variations in telomere length in humans is determined by inherited, telomere-near factors and is maintained throughout life. *Mech Ageing Dev* **124**, 629-640 (2003).
21. Doughty, B.R. et al. Single-molecule chromatin configurations link transcription factor binding to expression in human cells. *bioRxiv* (2024).
22. Jha, A. et al. DNA-m6A calling and integrated long-read epigenetic and genetic analysis with fibertools. *bioRxiv* (2023).

Reviewer #1 (Remarks on code availability):

NA

Reviewer #2 (Remarks to the Author):

The authors have significantly improved upon the manuscript and addressed most of the points that I had raised in the revised version of the manuscript. However, I have one lingering concern to address.

One of the key issues I previously raised was regarding the ability of the author's approach to accurately assign telomeres to the correct chromosomal arm in the absence of a reference genome. While I acknowledge the thorough discussion provided by the authors in the revised manuscript, as well as the inclusion of this concern in the figure legends, it remains imperative that this aspect is clearly highlighted in the figures depicting arm-level telomere length without a matched reference genome (e.g. Figure 2f and Extended Figures 3c, 4c, 5c, 6d). This is because a busy reader will very likely overlook this issue and will very likely misinterpret data presented in these figures. As such, I would greatly appreciate it if the authors could specifically point out in these figures the chromosomal arms that they think have reliable arm assignments vs. those that do not (e.g. by highlighting arms with excessively high or low coverage as unreliable).

This is particularly important because a central aspect and attraction of Telo-seq lies in its capability to investigate telomere length at the chromosome level. Therefore, it's essential for readers to comprehend the potential limitations of the approach when applying the method to their work.

Reviewer #3 (Remarks to the Author):

The authors have addressed all my concerns and I have no other comments except to update their code release page and also the raw data that was used for their analysis.

For example, on their GitHub page, it still says:

TODO:

- Add contents table
- Add installation instructions
- Add usage instructions
- Add example config file
- Add example scripts
- Add example data
- Add example output

Response to Referees, NCOMMS-23-51397-T.

Reviewer #1 (Remarks to the Author):

Tobias T et al have demonstrated the capability to measure telomere length using Oxford Nanopore long-read sequencing platform. The authors also design a method (Telo-seq) to enrich for telomere containing genomic DNA, which resulted in more than 75-fold increase in telomeric read output comparing to whole genome sequencing. A customized Bonito telomere model was developed for basecalling and sequencing data analysis, which should be useful for scientists studying telomere length in cancer and aging. Allele-specific telomere length can be measured for all 44 chromosomal ends excluding the sex-determinant chromosomes, showing potential chromosome arm-specific length heterogeneity. In addition to these findings, the authors also apply this new telomere sequencing method to successfully differentiate TERT+ cancer cells and ALT+ cancer cells by measuring the dispersion of their telomere length distribution.

Overall, the results are clear, and the method is interesting. However, the current manuscript is missing key details for comprehensive evaluation of the data, which may limit this new approach to recover accurate mean telomere lengths as well as allele-specific telomere lengths.

Response: We thank the Referee for their support and constructive comments, which are making this a better manuscript. Please find detailed answers to the individual points of criticism below.

Major comments:

1. The Telo-seq method adapted a similar strategy as STELA by ligating telorette-based telomere adapters to the C strand of telomeric ends. This will inadvertently result in loss of information for telomere G tail overhangs, which is known to vary from 100-300 nucleotides in length in human cells. Therefore, the newly established Telo-seq may result in underestimation of actual telomere length. Given the peripheral blood mononuclear cells telomere shortening rate is about 40-50 nts per year, whether this method can be utilized for accurate measurement of telomere length shortening rate in general population remains to be validated.

Response: We agree with the reviewer that Telo-seq uses like STELA ¹ and TESLA ² telorette-based telomere adapters and that Telo-seq is therefore not providing any information about the telomeric G-rich overhang length. The same argument is true for STELA and TESLA, which are both well-established methods in the telomere field to measure telomere length. Thus, if the telomeric G-rich overhang length is critical for a specific research question, alternative methods to STELA, TESLA and Telo-seq have to be applied.

The telomere shortening rate of ~30-50 nt in adult peripheral blood mononuclear cells have been determined based on a larger cohort of individuals. First, the individual's telomere length was determined with TRF ³⁻⁵ or flow FISH ⁶, for example. Second, the determined telomere length was plotted against the donor age and linear regression analysis performed. Our data on IMR90 ^{E6E7} fibroblasts at different population doublings suggest that Telo-seq can differentiate telomere length in samples 5 population doublings apart (Fig. 2). Given that Telo-seq has similar or higher

resolution than TRF and age-dependent Telo-seq-based telomere length decrease is also observed in the fibroblasts of the investigated aging cohort (Fig. 4c), we therefore postulate that, comparable to previous work, an averaged peripheral blood mononuclear cell telomere shortening rate per year from a cohort can be determined with Telo-seq.

Reviewer 1 Comment: Based on previous estimation: in healthy human cells, telomeres are typically 3 to 15 kilobase pairs (kb) in length. They shorten at rates of 50–200 base pairs per replication,

and undergo 30–60 population doublings before becoming senescent⁷. While the authors have indicated that “Telo-seq can differentiate telomere length in samples 5 population doublings apart (Fig. 2)”, the difference in telomere length between samples that are 5 population doubling apart is about 200 nt based on the authors’ own estimation of 39 bp per doubling, which is equivalent to 5 year worth of telomere shortening in human peripheral blood mononuclear cells. While I have no doubt that the current Telo-seq has achieved higher resolution than TRF in bulk telomere length measurement, I am wondering how the authors have the confidence to differentiate the annual telomere shortening rate that is merely ~30-50 nt in adult peripheral blood mononuclear cells, which is about the shortening rate of one doubling in IMR90E6E7 cells.

Response to comment: We appreciate the reviewer’s comment that Telo-seq achieves higher resolution than TRF in bulk telomere length determination. In this manuscript we are measuring telomere length of *in vitro* cultured IMR90^{E6E7} fibroblast cells and calculate bulk telomere shortening rate per population doubling. Telomere length measurements of peripheral blood mononuclear cells (PBMCs) and their age-dependent, *in vivo* telomere shortening is not subject of this manuscript. Given that we have not investigated PBMCs, that fibroblasts and PBMCs are different cell types and grow and age differently *in vitro* or *in vivo*, we do not feel confident in speculating on the resolution of Telo-seq of aged PBMC samples.

Reviewer 1 Comment: Further, as shown in Supplementary Table 4, the number of reads in about 40% (107 out of 264) alleles is below 250. The telomere length estimated in these alleles will have much larger standard error (>100bp) as indicated in the simulation study in a recent paper measuring telomere length using Nanopore long-read sequencing platform⁸. Therefore, the standard error is much larger than the estimated telomere shortening rate by the authors, which is about 39bp per doubling. Hence, it is impossible to get accurate allele-specific telomere shortening rate using this set of data considering the limited signal to noise ratio (39bp to 100bp in 40% allele groups).

Response to comment: The shortening rate of 39 bp / PD is based on bulk telomere length distributions (Fig. 2e). All samples included in the bulk telomere length shortening rate analysis had at least 13135 telomeric reads (Fig. 2b). Thus, the standard error on the bulk shortening rate is very low (Supplementary Fig. 1h).

Reviewer 1 Comment: The lack of ability to accurately measure the G rich overhang brings more doubt for the accuracy of current method. The lack of accuracy for allele-specific telomere length using STELA and TESLA was well known.

Response to comment: In the manuscript we are not measuring the length of the G-rich overhang. Work from Sanchez et al¹, Karimian et al², Smoom et al³ and ours are all using telomeric probes against the G-rich overhang to specifically label telomeres prior nanopore sequencing. Additionally, Tham et al⁴ describe a library preparation method for PacBio sequencing using telomeric probes. None of the manuscripts measures the overhang length.

Reviewer 1 Comment: By the way, results from Sanchez, S.E. et al and Karimian, K. et al has adopted a different ligation method to enable the sequence of the G tail overhangs using Nanopore long-read sequencing platform. Whether there is difference in the accuracy of telomere length measurement using these two different methods remains to be addressed.

Response to comment: We are aware of both interesting complementary studies^{1,2}. We have co-submitted our manuscript together with Sanchez et al¹ and come to consistent results for HG002 control cells independently of the library preparation.

2. The customized Bonito telomere model that was developed for basecalling in Oxford Nanopore sequencing data analysis will be valuable to the research interest of telomere length in cancer and aging. Please explain the detailed training process for this new model. Please also describe the detailed normalization steps that have been applied to read coverage.

Response: We thank the reviewer for highlighting this point. We have expanded the method section in the revised manuscript.

3. Please show the raw read coverage in all figures, which is essential to judge the accuracy of telomere length measurement.

Response: We appreciate the reviewer's comment. In the revised manuscript, we have added the number of telomeric reads corresponding to each bulk telomere length analysis figure panel. A summary for each flow cell and sample including number of telomeric reads is shown in the Supplementary Table 1, 3, 7, 8, 10, 11. Due to figure font size limitations we have added additional Supplementary Tables 2, 4 5, 7, and 9 to the revised manuscript including detailed information about chromosome arm-specific telomere length analysis, including number of telomeric reads per chromosome arm. The revised figure legends link to the corresponding Supplementary tables.

Reviewer 1 Comment: If the font size limitation is the concern, the read depth can be included in the legend.

Response to comment: We provided detailed description per chromosome arm in the linked Supplementary Tables and Supplementary Data. Adding the read numbers in the figure legend would be incompatible with Nature communication's figure legend limit of 350 words per legend. For example, adding the number of reads per sample and chromosome arm would result alone in 264 additional words/numbers in the figure legend for Fig. 2f with six IMR90^{E6E7} samples with 44 chromosome arms, excluding any annotation.

4. The telomere length estimated from HG002 Chr. 21p before haplotype phasing is above 9kb (Fig. 1e and lines 107-108). However, the lengths were estimated to be 4-5kb and 10-11kb for maternal and paternal alleles separately (Extended Data Fig. 4a). So, if the two alleles have been sequenced in similar depth, the telomere length of Chr. 21p should be much smaller than 9kb. Chr.1p has the same problem. Could you elaborate more regarding these differences. In addition, the allele-specific telomere lengths for paternal Chr. 13p and Chr. 22p are missing in Extended Data Fig. 4a as well.

Response: The reviewer's assumption that the differences between the chromosome arm-specific telomere length of HG002 and the two corresponding alleles can be due to different allelic coverages (Supplementary Table 2 and 5) is correct. This is likely the reason why the 1p chromosome arm-specific telomere length is closer to the maternal allele-specific telomere length (610 maternal and 337 paternal reads) (Supplementary Table 2 and 5). For Chr. 21p, both alleles have roughly the same number of reads (133 maternal and 113 paternal). However, we like to note that the boxplots show the median telomere length for each chromosome arm. Therefore, in the cases, in which the maternal and paternal telomere length are substantially different from each other and form distinct telomere length populations, even with same allelic coverage, the median will be closer to one allelic population than to the other one and likely be different than the mean telomere length of both populations. As suggested by the reviewer in major comment 3, we have added the underlying number of reads to Supplementary Table in the revised manuscript.

The paternal HG002 alleles of chromosome arm 13p and 22p are not shown in Fig. 3, because the mapping quality of both paternal alleles are below the applied mapping quality filter of 5. We have added a comment in the figure legend to explain why only the telomere length of the maternal 13p and 22p alleles are shown. We have also added a plot of the mapping quality per allele in HG002 in Supplementary Fig. 4a of the revised manuscript.

Reviewer 1 Comment: One of the major claims of the paper is "allele-specific human telomere lengths using Oxford Nanopore Technologies' native long-read sequencing." If the method failed to provide the allele-specific length for all chromosomal ends, then the authors should not make the claim.

In addition, no information is provided for the X and Y chromosomal ends for HG002.

Response to comment: We state that chromosome arm-specific telomere length can be addressed with Telo-seq, and only in certain scenario's (with best e.g. phased T2T reference genome available, like HG002) allele-specific telomere length. We address the limitations of chromosome arm and allele-specific telomere length extensively in the discussion. For HG002, similar allele-specific telomere length results were presented by Karimian et al² in an independent study, suggesting that telomere sequencing can indeed be used to resolve allele-specific telomere length.

5. In Extended Fig. 4a, it is very clear that both paternal and maternal allele in HG002 chr18q have two sets of reads in it: One is about 4-6kb, the other is about 8-10kb. Why the two alleles in this specific arm exhibit 2 sets of telomeres with different length?

Response: We agree with the reviewer that for both HG002 chromosome 18q alleles the majority of reads cluster around 8-10 kb and there is an accumulation of reads at 4-6 kb. The appearance of shorter telomeres could be indicative of a subpopulation of cells that have acquired shorter telomeres at chromosome 18q, for example due to rapid telomere shortening. The relatively long telomere length of 4-6 kb will likely not interfere with growth as other HG002 chromosome arms have shorter telomere length.

Comment: Any evidence to support the claim that “The appearance of shorter telomeres could be indicative of a subpopulation of cells that have acquired shorter telomeres at chromosome 18q, for example due to rapid telomere shortening.”

Alternative explanation is the two alleles from 18q have different telomere length, and the difference in abundance is due to difference in read depth of the two alleles. Same problem throughout the study.

Response to comment: As the read depth of 18q maternal (571 reads) and paternal (576 reads) is very similar, the alternative explanation provided by the referee is unlikely.

6. It is known that telomere length is reversely correlated with age. However, the authors have identified an 82-year-old individual who has abnormal longer telomere compared to the other individuals in his/her age group (Fig. 3a). How many replicates have been sequenced using Oxford Nanopore platform for each individual? Have the authors compared using the traditional TRF method to measure the telomere length in these eight individuals to validate the accuracy of their telomere length estimation using Oxford Nanopore platform.

Response: The reviewer stresses an important point with this comment. Previous work has shown that the telomere length of individuals in the population even in the same age group is very heterogenous ^{6, 10, 11}. Whereas all of these studies observe a reverse correlation of telomere length with age, there are also always individuals reported that have abnormal long telomeres relative to their respective age group, but also in comparison to individuals several decades younger. Therefore, we think that the 82-year-old individual present in our small aging cohort is indeed one of these individuals, who has relatively long telomeres for their age.

As we have shown reproducibility of Telo-seq results for HG002 and cancer samples (Supplementary Table 1, 11), we have sequenced one replicate per sample for the aging cohort samples (Supplementary Table 6). We have compared TRF and Telo-seq telomere length for the HG002, IMR90 ^{E6E7} and cancer samples and got overall consistent results between both methods (Fig. 1, 2, 5). Due to limited access to sample, we could not run a TRF on the eight individuals in the aging cohort.

Reviewer 1 Comment: Additional replicate or data from alternative method is necessary to confirm the accuracy of the telomere length measurement using the Telo-seq.

Response to comment: We have shown that the bulk telomere length assessment is reliable and comparable to TRF. In addition to our manuscript, there are now four preprints or manuscripts¹⁻⁴ online or published that use different, but similar protocols in combination with long-read sequencing. All manuscripts have in common that they initially compare the novel telomere length measurement method to other well-established telomere length measurement protocols and then continue exclusively with the newly developed method.

7. While Telo-seq may be able to distinguish TERT+ cancer cells and ALT+ cancer cells, the telomere length distribution in ALT cells obtained from Nanopore sequencing seem to be much shorter than the traditional teloblot (compare Fig. 4b and Fig. 4a). What is the potential cause for such discrepancy? The intense high molecular weight fractions in ALT samples seem to indicate incomplete genomic DNA digestion as well.

Response: We agree with the reviewer that the telomere length distribution of the ALT+ cancer cells seem to be longer in the TRF analysis than in Telo-seq. Based on the methodical differences there are several explanations:

- 1) For the TRF analysis, high-molecular weight genomic DNA is digested with restriction enzymes (AluI/MboI in our protocol) that frequently cut outside the telomeric motif. So telomeric TRF fragments contain telomeric repeats and adjacent subtelomeric sequence. Therefore, the detected telomere length in TRF is the sum of the telomere and the remaining subtelomere present in the fragment, whereas for Telo-seq we are only plotting the telomere length. Longer estimated telomere length based on TRF relative to telomere length determined by long-read sequencing have been also reported in a recent study for mouse cells³.

Reviewer 1 Comment: If this is the case, it is certainly possible to estimate the telomere length based on the Alu/MboI sites that can be identified in the long-read sequencing data from Nanopore. Please provide the data.

Response to comment: The reviewer is correct that Alu/MboI sites can be identified in long-read sequencing data. However, we do not see the biological relevance in providing this additional data.

- 2) TRF uses a telomeric probe to stain telomeric repeat containing fragments. As consequence, fragments with more telomeric repeats (longer telomeres) will allow more probe binding, resulting in higher intensity for longer than for shorter telomeres. For example, HT-29 and SK-N-AS have more intense telomere distribution than Calu-3 (Fig. 5a). Thus, shorter telomeres appear weaker on a TRF than longer telomeres. Contrary, in Telo-seq plots every telomeric read will be equally visualized, independently of its length.

Comment Referee 1: This is certainly more reasonable. But it does not explain the drastic difference in telomere length shown in TRF and Telo-seq.

Response to comment: We have provided an explanation in the discussion.

- 3) Fragments in a TRF analysis are separated on an agarose gel which runs non-linear. Smaller fragments will be separated on a larger distance. Consequently, longer fragments run closer to each other and are relatively compressed compared to shorter fragments. Thus, in addition to the probe binding, the gel running behavior is contributing to the visual impression of higher fragments being more abundant than shorter fragments, especially in samples with very heterogenous telomere length distributions like ALT⁺ cancer cells.

Comment Referee1: Large fragment will also have less efficiency to transfer as well.

Response to comment: We agree with the reviewer that larger fragments transfer less efficiently. Thus, to improve transfer of larger telomeric fragments to the membrane in our TRF protocol, the separated DNA was depurinated with HCl prior to blotting (please see Methods for details).

- 4) In TRF, the probe is staining any fragment which contains telomeric repeats, including interstitial telomeric repeats. These interstitial telomeric repeats can be generated in a chromosome fusion event and will lead to distinct bands in a TRF, for example. In contrast, Telo-seq captures the telomeric ends by using telorette-based adapters specific for the telomeric overhang and is therefore very unlikely to sequence these interstitial telomeric repeats. In highly rearranged and genetically unstable genomes like cancer genomes, especially ALT⁺ cancer cells, these interstitial telomeric repeats may be more common.

Comment Referee 1: Any actual evidence to support such assumption?

Response to comment: Cancer cells are more genomic instable than non-transformed, human cells⁵. Extensive genomic rearrangements have been previously observed in ALT⁺ cancer and have been suggested to be a consequence of the telomere dysfunction prior immortalization⁶.

We anticipate that with the advances of long-read sequencing technologies, cancer genome assemblies will become in large more achievable and available. Hence, the nature, length, and location of interstitial telomeric repeats in these rearranged genomes will be more apparent in future.

Taken together, we like to argue that these methodological differences contribute to and explain the impression that ALT cells seem to have longer telomeres based on TRF analysis.

Incomplete digestion of genomic DNA could be an explanation for high molecular weight fraction. However, based on the SybrGold staining that we have done for the gel prior to Southern blotting (Reviewer Fig. 1), we do not think that the high molecular weight fragments are due to incomplete digestion.

Reviewer Fig. 1. TRF analysis of cancer cell lines. AluI/MboI digested genomic DNA of 5 TERT⁺ and ALT⁺ cancer cell lines were separated on a 0.7% agarose gel. Left) agarose gel stained with Sybr Gold (1:10,000 in 0.5x TBE, 40 min, RT). Right) TRF using TelG telomeric probe (as shown in Fig. 5a)

Comment Referee 1: Looking at the SybrGold staining, it appears to have high molecular weight fragments at the very top of each lane on the gel. Please provide SybrGold staining gel picture with increased contrast.

Response to comment: Please see below the SybrGold staining with increased contrast (Updated Reviewer Fig. 1). There is no unusual accumulation of large fragments apparent.

Updated Reviewer Fig. 1. TRF analysis of cancer cell lines. AluI/MboI digested genomic DNA of 5 TERT⁺ and ALT⁺ cancer cell lines were separated on a 0.7% agarose gel. Left) agarose gel stained with Sybr Gold (1:10,000 in 0.5x TBE, 40 min, RT, overexposed). Right) TRF using TelG telomeric probe (as shown in Fig. 5a)

8. The author indicated that “In ALT+ cells, we could measure telomeres from 47 bp to 134.7 kb in length.” The 47 bp telomere seem to be too short. How often does the sample show telomere reads with such short length (percentage)?

It is highly possible that such fragments (47 bp) may be resulted from random shearing of genomic DNA during genomic DNA extraction.

Response: The very short telomere lengths, below 50 and 100 bp, are generally very infrequent (Reviewer Fig. 2). Only the ALT⁺ cell line SK-LU-1 with the shortest telomere length distribution had a significant accumulation of very short telomeres below 100 bp (~4%) (Reviewer Fig. 2 c). Telomeres between 1-50 bp are very infrequent (Reviewer Fig. 2d).

We cannot completely rule out that some of the reads with very short telomeres are due to random sharing during DNA prep or handling. However, as we are using a probe against the overhang to introduce the sequencing adapter, we consider it as rather unlikely that the randomly

shared DNA will produce an overhang that is compatible with the probe. Further, as postulated for cells without an active telomere maintenance mechanism, short telomeres (<1 kb) are accumulating in IMR90^{E6E7} with increasing population doubling (Fig. 2d, Reviewer Fig. 2), suggesting that the short telomeres are present in these samples and not (exclusively) a product of sample preparation (Reviewer Fig. 2). Finally, independently to us, the Artandi lab also reports the presence of short telomeres in their preprint on human telomere nanopore sequencing (<https://doi.org/10.1101/2023.11.29.569263>).

Reviewer Fig. 2. Distribution of telomere length in cancer and IMR90^{E6E7}. **a**, binned telomere length in base pairs (bp) of cancer cell lines. **b**, binned telomere length IMR90^{E6E7} at indicated population doublings (PD). **c**, bar graph of the percentage of telomeres below 100 bp. Number of telomeric reads with a telomere length between 1-100 bp is shown above the bar. **d**, bar graph percentage of telomeres below 50 bp. Number of telomeric reads with a telomere length between 1-50 bp is shown above the bar.

Comment Referee 1: Based on Reviewer Fig. 2, telomeres with 1-100 bp exist even in IMR90E6E7 that is in earlier passage of 66.2. Such results will indicate substantial induction of telomere dysfunctional foci in IMR90E6E7 that is in earlier passage. Has the author done immunohistochemistry to confirm such finding? I can foresee chromosomal ends with extremely short telomere do exist. However, one will expect such extremely short telomeres will increase

with passage in vitro. Based on Reviewer Fig. 2 above, the fraction of telomeres with 1-100 bp seem to be constant during in vitro passage. One will expect such fraction increases with in vitro passages as the telomeres that are 101-500 bp in length that is shown in above Fig. 2b. Can the authors provide explanation for such discrepancy for telomeres that are 1-100 bp and 101-500 bp?

Response to comment: As shown in Reviewer Fig. 2c, IMR90^{E6E7} telomeres between 1-100 nt in length are increasing from 0.18% at PD66.2 (youngest IMR90^{E6E7} cells) to 0.41% at PD106.1 (oldest IMR90^{E6E7} cells) of all reads. Thus, in line with general telomere shortening, both the fraction of the short telomeres between 1-100 and 101-500 bp are accumulating in IMR90^{E6E7} with increasing population doublings.

9. The Allele-specific telomere length measurement is certainly one of the most interesting results in this manuscript.

Response: We thank the reviewer for this comment that encouraged us to move the mapped phased HG002 telomere length (resubmitted manuscript Supplementary Fig. 4a) as independent main text Figure 3 in the revised manuscript. We moved the paragraph about allele specific telomere length assignment up in the revised manuscript text right after the “Telo-seq resolves telomere shortening” paragraph.

While the authors showcase the capability to measure the telomere length of all 88 alleles, whether they are accurate remains a major question. For example: While the author highlighted their ability to accurately measure telomere shortening rate (average 39 bp per PD) in IMR90E6E7 cells in serial passage in vitro, the allele-specific telomere length shortening seems to have a lot of heterogeneity. In many instances (Fig. 2f, Chr. 22p PD106.1; Chr. 14p PD106.1; Chr 12q PD106.1, etc) the allele-specific telomere lengths in the late passage are much longer than IMR90E6E7 in earlier passages. These discrepancies of allele-specific telomere length bring a lot of doubt on the accuracy of allele-specific telomere length measurement using current method.

Response: As the reviewer has correctly pointed out, the median chromosome arm-specific telomere length is increasing in some chromosome arms especially in the later passages (PD106). However, we like to note that these cells (IMR90^{E6E7} PD106) are encountering replicative crisis⁷⁻⁹. Crisis, or mortality stage 2, is a telomere-dependent proliferation barrier, in which most cells die an autophagy-dependent, innate immunity-driven cell death^{10, 11}. Thus, cells with dysfunctional telomeres will be efficiently removed from the population. As telomere length in a cellular population is distributed around an averaged telomere length¹², there will be some cells with overall shorter, or with short telomeres, at only some chromosome arms. We postulate that this subpopulation of cells will experience crisis earlier than the rest of the population. Thus, our interpretation of the results is that there is counterselection against the subpopulations of cells that have very short bulk telomeres, or very short telomeres, at individual chromosome arms in cells approaching crisis as they are experiencing crisis earlier and are removed from the population. Consequently, under these selection conditions, clones with longer telomeres will

outgrow the ones with shorter ones. For the last sampling point (PD106), but likely also for cells that bypass the senescence plateau (PD96), there are two processes affecting the chromosome arm-specific telomere length measurements: telomere shortening and selection for a population of cells with telomeres long enough to protect chromosome ends. Consequently, we have excluded the two last timepoints for the calculation of chromosome arm-specific shortening rates (Supplementary Fig. 3c).

In the revised manuscript we have included additional benchmarking figures based on HG002 Telo-seq analysis (Supplementary Fig. 1 and 2), updated the chromosome arm-specific telomere length figures using a mapping quality of 20 and discussed current limitations of chromosome arm-specific telomere length assignment in the “Discussion”.

Comment Referee 1: If the authors’ speculation that “Consequently, under these selection conditions, clones with longer telomeres will outgrow the ones with shorter ones.” is valid. One will expect this should be true for all the chromosomal ends at PD106.1. However, this is certainly not the case.

Similar arguments apply to many of the allele-specific telomere length estimation using reads that are less than 20 in total (Supplementary Table 4_Telo-seq). Are these estimations accurate? Since the authors did not provide additional data to rectify their conclusion, doubts remain on the accuracy of allele-specific telomere length measurement using Telo-seq at current state.

Response to comment: We agree with the reviewer and showed that the accuracy of the mean/median telomere length is dependent on number of reads (Supplementary Fig. 1h). We discuss limitations of chromosome arm- and allele-specific telomere length in the manuscript’s discussion and now state stronger that the mean/median telomere length is dependent on the number of underlying reads.

In addition, only the chromosomal end specific telomere lengths are presented, but not the allele-specific telomere lengths of IMR90E6E7 cells are shown for each PD? Given the differences of allele-specific telomere length (Extended Data Fig. 4c), plotting the allele-specific telomere shortening rate in Fig. 2f will be important.

Response: We agree with the reviewer that allele-specific shortening rates would be very interesting. However, resolving allele-specific telomere length is challenging and, given the heterogeneity, requires a high coverage per chromosome arm. Phasing quality also depends on the reference genome. We present a de novo approach (Supplementary Data Fig. 4b,c); however, as discussed in the manuscript, this approach is not optimal, and we cannot resolve all alleles for every chromosome arm. One key issue is that for IMR90^{E6E7} as well as for all other samples in this study, except for HG002, there is no phased reference genome and parental genome information available. Given these challenges we do not feel confident in calculating allele-specific shortening rate at this point.

Comment Referee 1: If this is the case, the authors should not claim to achieve “allele-specific human telomere lengths using Oxford Nanopore Technologies’ native long-read sequencing” in general.

Response to comment: In line with work from Karimian et al², allele-specific human telomere length is achievable for samples with matched phased reference T2T genome. We suggest a de novo phasing approach for samples without phased reference genome and discuss limitations.

10. By profiling allele-specific telomere length using donor-derived fibroblasts, the authors have indicated that some chromosome arms consistently had shorter or longer telomeres. They further proposed that there are potential conserved chromosome arm-specific features influence telomere length. Similar results have been reported previously (Londono-Vallejo J.A. NAR 2001; Martens U.M. Nature Gen. 1998; Graakjaer J. Mech. Ageing Dev. 2003 etc) using other telomere length measurement methods, such as FISH. However, there are no consistency on which chromosomal end has the shortest or the longest telomere.

Response: We thank the reviewer for this comment. Given the high heterogeneity of the telomere length on the chromosome arm level and the relatively small cohort size (≤ 20 individuals in previous^{20, 21} and 9 in our work), we believe that to comprehensively address the question whether specific telomeres at some chromosome arms are consistently shorter or longer requires a large, diverse, and well-controlled cohort. Nevertheless, despite some discrepancy with and within previous data, the five shortest (chromosome 21p, 19p, 19q, 16q, 9q) and longest telomeres (chromosome 18q, 12q, 3p, 4q, 5p) identified in our cohort are also detected to be shorter and longer in two previous studies^{20, 21}. We have expanded this point in the discussion of the revised manuscript.

Similar results were observed in Extended Data Fig. 4a: most of the two alleles from the same chromosomal end seems to have vast different telomere length. The allele-specific telomere length for the eight samples should be shown in Extended data Fig. 3c. In summary, whether there is conserved chromosome arm-specific features influence telomere length remains to be addressed.

Response: Telomere phasing without the sample's phased reference genome is difficult and our de novo phasing approach frequently results in alleles and chromosome arms that cannot be resolved (Supplementary Fig. 4b,c). In addition, we do unfortunately not have enough coverage to present allele-specific telomere length for the eight fibroblasts.

We agree with the reviewer that it is an open question, whether conserved chromosome arm-specific features exist that influence human telomere length. Our analysis is limited to only 9 fibroblast samples. Whether there are chromosome arm-specific features that influence telomere length, and which chromosome arms have the longest/shortest telomeres has to be determined in future studies. To answer these questions comprehensively, analysis of a much larger, well-controlled, diverse cohort is required. This is beyond the scope of this manuscript.

11. What is the meaning of the methylation data? There is a lot of heterogeneity, which may be due to the sequencing method itself or read depth. Is the methylation results shown in each panel in extended data Fig. 6 average of multiple reads from the same allele? The results have shown

that ALT+ cancer cell lines were observed to be hypomethylated compared to TERT+ cancer cell lines. However, it is also mentioned that not all the arms were equally hypomethylated. Can the authors specifically list out the hypomethylated/hypermethylated chromosome arms? And within the differentially methylated arms, what are the differentially methylated regions (DMR)? Does the ALT+ cell lines have similar DMR compared to TERT+ cell lines? Furthermore, it is also important to compare the methylation pattern of cancer cells with normal cells (HG002 and IMR90).

Response: The advantage of native nanopore sequencing and Telo-seq is that, in addition to the sequence information and telomere length, the methylation status of the CpG sites closest to the telomeres can be obtained. We agree there is inter- and intra-sample heterogeneity. However, to our knowledge this type of analysis has not been done in this detail before and highlights which type of questions can be answered by Telo-seq.

We thank the reviewer for suggesting a summary about the methylation status of the CpG sites closest to the telomeres in controls and cancer cell lines. This will provide a fast overview and facilitate the search for the chromosome arms where the methylation status differs from controls. We have generated a heatmap that shows the averaged methylation of every chromosome arm for the indicated sample and added it to Supplementary Fig. 7.

Comment Referee 1: I would like to point out that there are two recent papers reporting the DNA methylation profiling at telomeres.

Response to comment: We agree with the reviewer that obtaining methylation information in addition to genetic sequence is an exciting piece of additional data that can be obtained by long-read sequencing technologies. In our opinion, the presence of additional recent preprints reporting DNA methylation profiling at telomeres highlights the interest in this additional epigenetic information that can be obtained with nanopore sequencing and Telo-seq.

Reviewer #2 (Remarks to the Author):

The authors have significantly improved upon the manuscript and addressed most of the points that I had raised in the revised version of the manuscript. However, I have one lingering concern to address.

One of the key issues I previously raised was regarding the ability of the author's approach to accurately assign telomeres to the correct chromosomal arm in the absence of a reference genome. While I acknowledge the thorough discussion provided by the authors in the revised manuscript, as well as the inclusion of this concern in the figure legends, it remains imperative that this aspect is clearly highlighted in the figures depicting arm-level telomere length without a matched reference genome (e.g. Figure 2f and Extended Figures 3c, 4c, 5c, 6d). This is because a busy reader will very likely overlook this issue and will very likely misinterpret data presented in these figures. As such, I would greatly appreciate it if the authors could specifically point out in these figures the chromosomal arms that they think have reliable arm assignments vs. those that do not (e.g. by highlighting arms with excessively high or low coverage as unreliable).

This is particularly important because a central aspect and attraction of Telo-seq lies in its capability to investigate telomere length at the chromosome level. Therefore, it's essential for readers to comprehend the potential limitations of the approach when applying the method to their work.

Response: We thank the Referee for appreciating our efforts in highlighting current limitations in chromosome arm-specific/allele-specific telomere length assignments and their constructive revision comments, which are making this a better manuscript. According to the referee's suggestions, we have clearly and sufficiently indicated limitations in all relevant figure legends and the discussion.

Reviewer #3 (Remarks to the Author):

The authors have addressed all my concerns and I have no other comments except to update their code release page and also the raw data that was used for their analysis.

For example, on their GitHub page, it still says:

TODO:

Add contents table

Add installation instructions

Add usage instructions

Add example config file

Add example scripts

Add example data

Add example output

Response: We thank the reviewer for the supportive revision comments. We have updated the GitHub page.

References:

1. Sanchez, S.E. et al. Digital telomere measurement by long-read sequencing distinguishes healthy aging from disease. *bioRxiv* (2023).
2. Karimian, K. et al. Human telomere length is chromosome end-specific and conserved across individuals. *Science*, eado0431 (2024).
3. Smoom, R. et al. Telomouse—a mouse model with human-length telomeres generated by a single amino acid change in RTEL1. *Nat Commun* **14**, 6708 (2023).
4. Tham, C.Y. et al. High-throughput telomere length measurement at nucleotide resolution using the PacBio high fidelity sequencing platform. *Nat Commun* **14**, 281 (2023).
5. Hanahan, D. & Weinberg, R.A. Hallmarks of cancer: the next generation. *Cell* **144**, 646-674 (2011).
6. Lovejoy, C.A. et al. Loss of ATRX, genome instability, and an altered DNA damage response are hallmarks of the alternative lengthening of telomeres pathway. *PLoS Genet* **8**, e1002772 (2012).
7. Stein, G.H. SV40-transformed human fibroblasts: evidence for cellular aging in pre-crisis cells. *J Cell Physiol* **125**, 36-44 (1985).
8. Wright, W.E., Pereira-Smith, O.M. & Shay, J.W. Reversible cellular senescence: implications for immortalization of normal human diploid fibroblasts. *Mol Cell Biol* **9**, 3088-3092 (1989).
9. Shay, J.W., Pereira-Smith, O.M. & Wright, W.E. A role for both RB and p53 in the regulation of human cellular senescence. *Exp Cell Res* **196**, 33-39 (1991).
10. Nassour, J. et al. Telomere-to-mitochondria signalling by ZBP1 mediates replicative crisis. *Nature* **614**, 767-773 (2023).
11. Nassour, J. et al. Autophagic cell death restricts chromosomal instability during replicative crisis. *Nature* **565**, 659-663 (2019).
12. Rufer, N., Dragowska, W., Thornbury, G., Roosnek, E. & Lansdorp, P.M. Telomere length dynamics in human lymphocyte subpopulations measured by flow cytometry. *Nat Biotechnol* **16**, 743-747 (1998).